# Genetic code expansion reveals site-specific lactylation in living cells reshapes protein functions

Chang Shao[1,4], Shuo Tang[2,4], Siqin Yu[1,4], Chenguang Liu[1], Yueyang Zhang[1], Tianyan Wan[2], Zimeng He[1], Qi Yuan[1], Shihan Wu[2], Hanqing Zhang[1], Ning Wan [1], Mengru Zhan[2], Ren Xiang Tan [2,3], Haiping Hao [1], Hui Ye [1] ✉ & Nanxi Wang [2] ✉

Protein lactylation is an emerging field. To advance the exploration of its biological functions, here we develop a comprehensive workflow that integrates proteomics to identify lactylated sites, genetic code expansion (GCE) for the expression of site-specifically lactylated proteins in living cells, and an integrated functional analysis (IFA) platform to evaluate their biological effects. Using a combined wet-and-dry-lab proteomics strategy, we identify a conserved lactylation at ALDOA-K147, which we hypothesize plays a significant biological role. Expression of this site-specifically lactylated ALDOA in mammalian cells reveals that this modification not only inhibits enzymatic activity but also induces gain-of-function effects. These effects reshaped ALDOA functionality by enhancing protein stability, promoting nuclear translocation, regulating adhesion-related gene expression, altering cell morphology and modulating ALDOA-interacting proteins. Our findings highlight the utility of the GCE-based workflow in establishing causal relationships between specific lactylation events and both target-specific and cell-wide changes, advancing our understanding of protein lactylation's functional impact.

Protein lactylation is derived from lactate. It was first discovered on lysine residues of human histone proteins and was defined as an epigenetic mark that regulates gene expression in diverse biological contexts[1]. Recently, it was posited as a widespread post-translational modification (PTM) that also modifies non-histone proteins[2–8]. Using ALDOA as an example, we found that ALDOA conservatively carries lactylation at its active site, and therefore, its activity is postulated to be inhibited by this PTM[3]. These representative studies have attracted enormous attention to the conceivably essential role of lysine lactylation during recent years and solicit researchers to seek answers to elusive questions such as: can lactylation instruct functional changes on the modified non-histone proteins? Specifically, is lysine lactylation capable of inducing diverse biological consequences in living cells ranging from regulating modified protein's activity and stability to fine-tuning interactions and even navigating proteins to certain compartments or organelles? Does lactylation installed at distinct lysine residues differentially control modified proteins?

However, addressing these questions of intense biological interests is challenging because modification enzyme-based regulation is promiscuous and fails to site-specifically impose lactylation on the target protein in native biological contexts[9]. Furthermore, K to Q/E mutagenesis-based approaches[5,7,10,11] commonly used to mimic protein acetylation cannot fully mirror the lactylation state, nor can they distinguish between acetylation and lactylation

[1]Jiangsu Provincial Key Laboratory of Drug Metabolism and Pharmacokinetics, State Key Laboratory of Natural Medicines, China Pharmaceutical University, Tongjiaxiang No. 24, Nanjing, Jiangsu, China. [2]School of Pharmacy, Nanjing University of Chinese Medicine, Xianlindadao No. 138, Nanjing, Jiangsu, China. [3]State Key Laboratory of Pharmaceutical Biotechnology, Institute of Functional Biomolecules, School of Life Sciences, Nanjing University, Nanjing, Jiangsu, China. [4]These authors contributed equally: Chang Shao, Shuo Tang, Siqin Yu. ✉e-mail: cpuyehui@cpu.edu.cn; nanxi.wang@njucm.edu.cn

of target proteins. This calls for technical advances enabling precisely engineered lactylation. To address this need, we and others have used GCE to site-specifically incorporate lactylated lysine (Klac) into target proteins, such as green fluorescent protein (GFP) and luciferase, in living cells using Klac-specific tRNA synthetase (KlacRS)/tRNA$_{CUA}^{Pyl}$ pairs[3,8,12,13]. As this approach allows for convenient site-directed lactylation, we anticipate that it would be particularly suitable for inferring a causal link between lactylation at a specific site for a given protein of interest (POI) and its regulatory outcomes in a native and complex biological context (Fig. 1a). This allows better understanding of the biology of lactylation on non-histone proteins, considering previous studies have mostly interrogated functionality of histone lactylation[1,14–18].

In this work, we first established a wet-and-dry-lab proteomics approach to mine functionally essential lactylation (Fig. 1a). We find that, besides affinity-enriched lactylproteome data repeatedly pinpointing lysine lactylation at residue K147 of ALDOA (ALDOA-147Klac), this modification is consistently identified in public proteome data of multiple human cell lines and tissue types using our cyclic immonium (CycIm) ion-based data mining strategy[3]. ALDOA-147Klac also shows high occupancy in analyzed data and is even evolutionarily conserved in mammals and insects, suggesting its functional significance.

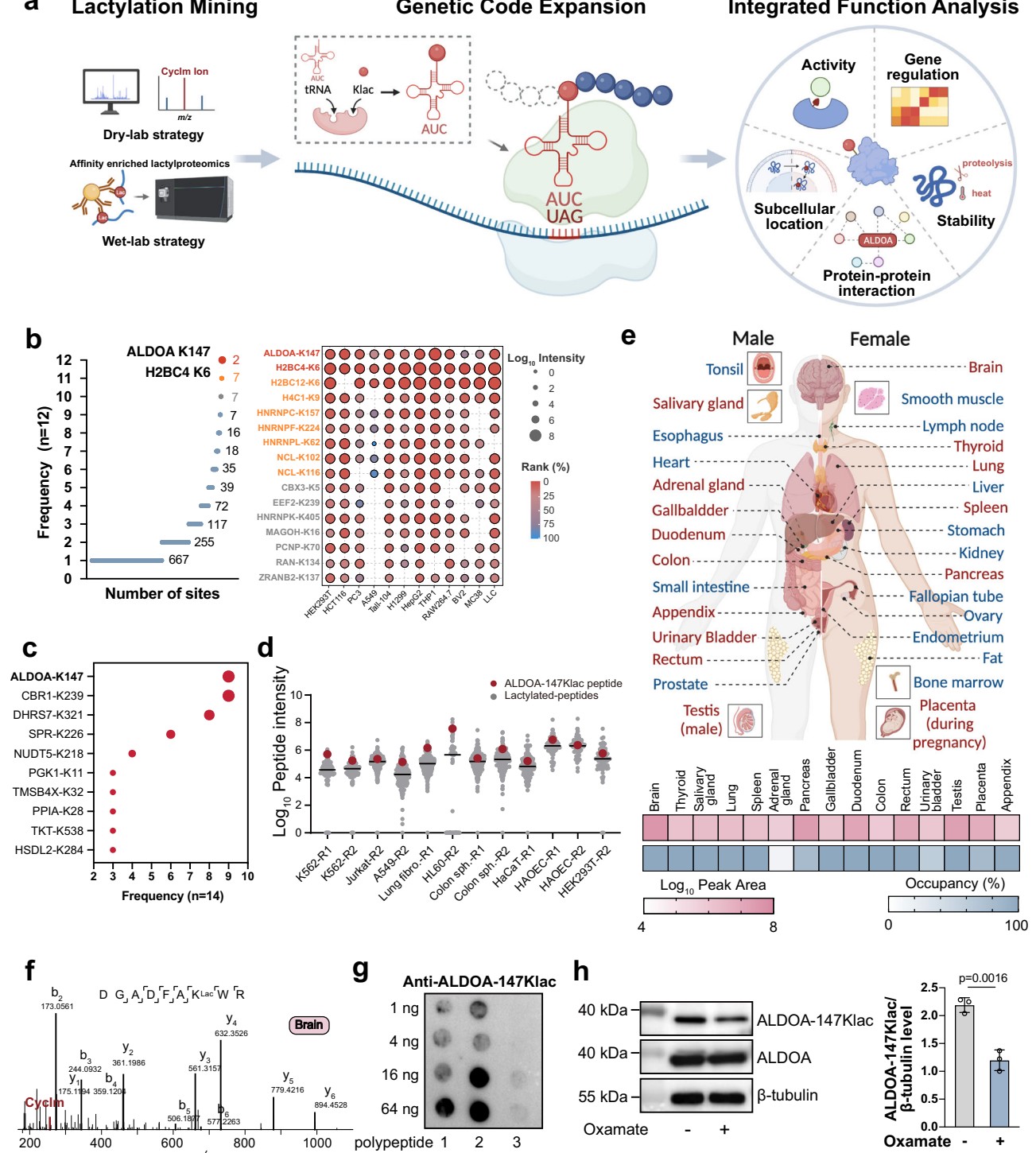

**Fig. 1 | Proteomics mining pinpoints functionally important lactylation on ALDOA. a** The workflow first uses proteomics to mine understudied lactylation sites of potential functional importance, then uses GCE to express site-specifically lactylated proteins in living cells, followed by assembling a suite of biochemical tools to assess the biological consequences of site-specific lactylation. Created in BioRender. Shao, C. (2024) BioRender.com/a84h467. **b** Detection frequency of the lactylation sites in the affinity-enriched lactylproteome of representative human and mouse cell lines (left panel). Cells were treated with 25 mM lactate for 24 h to stimulate lactylation. The ion intensity (bubble size) and ranking (bubble color) of the identified lactylated peptides with a detection frequency of more than 10 times were shown in the corresponding bubble plot (right panel). **c** Detection frequency of the lactylation sites in the re-analyzed 14 human cell types retrieved from the Meltome Atlas (PXD011929). **d** Abundance of the lactylated peptides (gray) in the re-analyzed human cell proteome retrieved from the Meltome Atlas (PXD011929). Peptides carrying ALDOA-147Klac are highlighted in red. R1 and R2 represent the

two biologically independent experiments we re-analyzed. **e** Abundance and occupancy of the peptides carrying lactylated ALDOA-K147 in 15 healthy tissues (red) retrieved from a deep proteome atlas of 29 human healthy tissues (PXD010154). Created in BioRender. Shao, C. (2024) BioRender.com/w02j352. Occupancy was estimated based on the ratio of the abundance of the lactylated peptides divided by the sum of the lactylated and non-lactylated peptides at this site. **f** Representative MS/MS spectrum with the signature CycIm ion revealing lactylation on ALDOA-K147 in human brain (PXD010154). **g** Dot blot assay of the generated ALDOA-147Klac antibody. Peptide 1 and 2, two synthesized ALDOA-147Klac-bearing peptides; peptide 3, an unmodified ALDOA-K147-bearing peptide. **h** Immunoblots of ALDOA-147Klac in HEK293T cells treated with oxamate (25 mM, 24 h). Data represent the mean ± S.D. ($n$ = 3 biologically independent samples) and the $p$ value was calculated by unpaired two-tailed Student's $t$ test. Source data are provided as a Source Data file.

With the developed GCE technique, we genetically engineered site-specifically lactylated ALDOA-147Klac in HEK293T cells. Then, we established an IFA platform for functional assessment of this specific lactylation and collected a wealth of ALDOA-centric information, including enzyme activity, stability, subcellular location, gene regulation and interactions (Fig. 1a). We discover that this single-site lactylation is able to completely reshape the function of ALDOA: it not only abolishes the enzyme activity of ALDOA and inhibits the glycolytic flux, but also enhances the protein thermal stability, navigates ALDOA to the nucleus, affects the transcriptome and modulates the interacting partners of ALDOA. Multiple tiers of biological changes are induced, expanding far beyond cellular metabolism. Moreover, the workflow presented, by integrating powerful proteomics and GCE approaches with IFA (Fig. 1a), paves the way to establish causal links between the relatively nascent PTM—lactylation—and the biological consequences in living cells.

## Results

### Proteomics mining pinpoints functionally important lactylation on ALDOA

Certain site-specific PTMs are prevalent in cells and serve as central players in core cellular processes[19,20]. These are exemplified by PTMs such as histone acetylation/methylation that control transcription[21] and phosphorylation on receptor tyrosine kinases that regulate cell growth and differentiation[22]. In contrast to these well-studied PTM states, lactylation has remained relatively understudied, and most functional studies have been devoted to histone lactylation[1,14–18]. We therefore set out to discover additional non-histone lactylation sites of biological significance.

We deployed a combined wet-and-dry-lab proteomics approach that first discovered affinity-enriched lactylation sites in a limited number of cell types (wet-lab strategy), followed by identifying the recurrent sites via mining large-scale public proteomic data resource collected from multiple human cell types and tissues (dry-lab strategy). Using this combined strategy, we identified lactylated peptides from 8 human and 4 mouse cell lines and analyzed their frequency of occurrence among the cells tested (Fig. 1b, Supplementary Data 1). Among the identified lactylated peptides, ALDOA-147Klac was detected with the highest frequency, up to 100%. Furthermore, its intensities were consistently among the highest in almost all of the assayed cell lines (Fig. 1b). Naturally, the high prevalence and intensity of ALDOA-147Klac implies its biological significance.

To confirm the prevalence of ALDOA-147Klac in the human proteome, we used the dry-lab experimentation strategy[3] to screen for true lactylated peptides in unenriched public human proteome data resource. The presence of the CycIm ion in MS/MS spectra of target-decoy database-searched lactylpeptides was set as the gold standard to signify true lactylation[3]. With this strategy, we re-analyzed lactylated peptides from the Meltome Atlas (PXD011929)[3,23], and noted that K147

of ALDOA was once again the most commonly lactylated residue in the detected repertoire (Fig. 1c). We ranked the intensities of identified lactylated peptides and confirmed high abundances of ALDOA-147Klac-bearing peptides (Fig. 1d). In agreement with the high intensities, lactylation on K147 is estimated to have an occupancy as high as 50.33% among the human cell lines examined[3] (Supplementary Fig. 1a). In addition to human cell lines, we investigated the occurrence of ALDOA-147Klac in human tissues. Re-analyzing a deep proteome atlas of 29 healthy human tissues (PXD010154)[24] showed that ALDOA-147Klac is present in 15 of the analyzed tissue types (Fig. 1e, Supplementary Fig. 1b, Supplementary Data 2). The MS/MS spectra retrieved for ALDOA-147Klac produced the CycIm ion, which is specific to intra-peptide lactylated lysine (Fig. 1f). Once again, this peptide exhibited both relatively high ion intensities and occupancy in human vital organs such as the brain, lungs and digestive system (Fig. 1e and Supplementary Data 3). Finally, we searched publicly available affinity-enriched lactylproteome datasets together with unenriched proteomes of different species across the kingdoms of life[25]. We found that the sequences of ALDOA-K147 and its lactylation are conserved in human, mouse (our enriched proteomic data), rabbit (the kingdoms of life, PXD014877)[25] and even insects, such as fruit fly (Meltome Atlas, PXD011929)[23] and western flower thrips (public lactylproteome, PXD030799)[26] (Supplementary Fig. 1c, d and Supplementary Data 4). We then constructed a phylogenetic tree based on the sequence alignment of ALDOA from these 5 species using MEGA X (Supplementary Fig. 1e). We found ALDOA of *Homo sapiens* has a relatively close relationship with that of *Oryctolagus cuniculus* and *Mus musculus*, while showing a relatively distant relationship with those of the insects *Drosophila melaganoster* and *Frankliniella occidentalis*. As high levels of conservation often indicate important functions, this lactylation event across different species has aroused our interests.

### ALDOA-147Klac can be regulated by dynamic lactate levels

We then asked whether this specific ALDOA-147Klac responds to modulated lactate levels. To facilitate the direct detection of ALDOA-147Klac levels, we have generated an antibody against ALDOA-147Klac, which we named anti-ALDOA-147Klac. The dot blot assay confirmed that this antibody specifically recognized the K147-lactylated peptides of ALDOA rather than the unmodified form (Fig. 1g). We first administered lactate to HEK293T cells to determine whether elevated lactate levels would stimulate ALDOA-147Klac. However, no significant increase was found, likely due to the inherent high level of ALDOA-147Klac in the cells (Supplementary Fig. 1f). We then treated cells with two inhibitors of the lactate-producing enzyme lactate dehydrogenase (LDHA), oxamate and FX-11. Significant decreases in intracellular lactate were observed after both treatments, and, as expected, ALDOA-147Klac decreased concomitantly (Fig. 1h and Supplementary Fig. 1g, h). Taken together, the prevalence of ALDOA-147Klac in humans and conservation across species, along with its responsiveness

to dynamic lactate levels, suggest this site-specific lactylation as a functional hostpot, prompting further investigation into its biological consequences.

## Introducing site-specific lactylation in living cells with genetic code expansion

Intrigued by the biological changes caused by ALDOA-147Klac, we proposed to use GCE to produce site-specifically lactylated ALDOA in living cells and pursue the answer. Previously, we have evolved a pyrrolysyl (Pyl)-tRNA synthetase (PylRS) that specifically recognizes Klac (Supplementary Fig. 2a), which we named KlacRS1[3], and managed to genetically encode and harvest lactylated target proteins in *E. coli*. In this work, we set out to engineer KlacRS with higher incorporation efficiency and stringency in order to introduce lactylation site-specifically into POIs in mammalian cells.

To achieve this goal, we first conducted positive selection with a focused *Methanosarcina mazei* PylRS (*Mm*PylRS) mutant library by completely randomizing residues Y306 and C348 to NNK (*N* = A/T/G or C, K = T or G). A hit with the Y306M and C348T mutations, exhibiting a Klac-dependent phenotype, was selected and named KlacRS2 (Fig. 2a). Meanwhile, inspired by the outstanding performance of chimeric tRNA synthetase/chimeric tRNA$_{CUA}^{Pyl}$ systems[27], we created two chimeric KlacRS mutants, namely chKlacRS-WT (wild type) and chKlacRS-IPYE (IPYE mutant) by fusing the tRNA$_{CUA}^{Pyl}$ binding domain of *Methanosarcina barkeri* pyrrolysyl-tRNA synthetase (*Mb*PylRS-TD) or its active mutated form with the KlacRS1 catalytic domain at the C-terminus (KlacRS1-CD), respectively (Fig. 2a). To assess the amber suppression ability of KlacRS variants described above, we co-expressed them with an enhanced GFP (EGFP) reporter carrying the Y39TAG mutation (EGFP-39TAG) in the presence or absence of Klac in *E. coli*. Interestingly, the results showed that, although the orthogonality of KlacRS2, chKlacRS-WT, and chKlacRS-IPYE was enhanced compared with KlacRS1, KlacRS1 exhibited the highest Klac incorporation efficiency (Fig. 2b).

Next, we examined the Klac-incorporating capacity of KlacRS1 in mammalian cells. We co-transfected HEK293T cells with a plasmid expressing KlacRS1/tRNA$_{CUA}^{Pyl}$ pair, and a dual-reporter plasmid encoding mCherry-TAG-EGFP gene. By quantifying the proportion of mCherry$^+$EGFP$^+$ cells (cells with Klac incorporation) among all mCherry$^+$ cells (total transfected cells), we can fairly assess the efficiency of Klac incorporation. First, we found that the proportion of mCherry$^+$ cells was consistent across the groups examined by flow cytometry (at ~60%), indicating a stable transfection efficiency in our experiments. Strong EGFP fluorescence was only detected in mCherry$^+$ cells when Klac was also supplied, suggesting successful Klac incorporation and high orthogonality using KlacRS1. We found that the fluorescence intensity of Klac-incorporated EGFP was higher at 48 h than at 24 h. Importantly, the proportion of mCherry$^+$EGFP$^+$ cells among mCherry$^+$ cells reached 50% at 48 h, indicating good Klac incorporation efficiency (Fig. 2c). We also evaluated KlacRS1 by fluorescence microscopy. Consistent with the flow cytometry results, negligible EGFP fluorescence signal was detected in mCherry$^+$ cells in the absence of Klac, whereas strong EGFP and mCherry fluorescence signals were observed simultaneously only in the presence of Klac (Fig. 2d). Similarly, the proportion of mCherry$^+$ EGFP$^+$ cells increased when the Klac treatment was extended from 24 h (~31%) to 48 h (~53%), and 48 h was used in the following experiments. Collectively, we demonstrate that KlacRS1 is able to site-specifically incorporate Klac into POIs with high efficiency and fidelity in both *E. coli* and mammalian cells, and was therefore used for the following experiments.

Next, we tested different Klac concentrations to optimize its incorporation efficiency in mammalian cells. Specifically, we measured EGFP fluorescence intensity for HEK293T cells expressing Klac-incorporated EGFP via the GCE approach when different concentrations of Klac were administered. Both fluorescence imaging and flow cytometry revealed that EGFP intensities increased in a dose-dependent manner as Klac concentration was increased from 1 to 5 mM, reaching a plateau at 5 mM (Supplementary Fig. 2b, c). Treatment with more than 5 mM Klac did not promote EGFP expression (Supplementary Fig. 2b, c), but resulted in significant cytotoxicity (Supplementary Fig. 2d). Therefore, 5 mM Klac was chosen for the following experiments. We also compared the incorporation efficiency of KlacRS1 with the chimeric KlacRS (chKlacRS-IPYE) in HEK293T cells and found KlacRS1 to be more efficient with Klac incorporation (Supplementary Fig. 2e), validating our findings in *E.coli* cells (Fig. 2b).

Using the optimized Klac incorporation system, we sought to site-specifically incorporate Klac into ALDOA in living cells via GCE. We co-transfected HEK293T cells with the KlacRS1/tRNA$_{CUA}^{Pyl}$ pair and ALDOA-147TAG plasmids. Immunoblotting analysis with the anti-ALDOA-147Klac antibody and anti-Flag tag antibody confirmed the successful expression of ALDOA-147Klac in cells. We showed that treating cells with the ALDOA-WT plasmid at 40% of the ALDOA-147TAG plasmid resulted in comparable expression levels of both ALDOA-WT and ALDOA-147Klac (Fig. 2e and Supplementary Fig. 3a). Notably, the ALDOA-147TAG-transfected group produced ~60-fold and ~30-fold higher levels of ALDOA-147Klac than the endogenous group and the ALDOA-WT group, respectively (Supplementary Fig. 3b).

Further, we immunoprecipitated cells expressing Flag-tagged ALDOA-WT and ALDOA-147Klac with anti-Flag antibody, followed by bottom-up proteomic analysis. Precise Klac incorporation at K147 of ALDOA was confirmed by detecting lactylated K147-bearing peptides only in cells genetically encoding Flag-tagged ALDOA-147Klac, but not in cells encoding ALDOA-WT (Fig. 2f). Consistently, the non-lactylated K147-bearing peptide was only detected in the ALDOA-WT-encoding cells, but not in ALDOA-147Klac-encoding cells (Supplementary Fig. 3c).

## Lactylation on ALDOA-K147 abolished enzyme activity and regulated glycolytic flux

Considering ALDOA is a metabolic enzyme responsible for the conversion of fructose 1,6-bisphosphate (FBP) into two triose phosphate, glyceraldehyde-3-phosphate (G3P) and dihydroxyacetone phosphate (DHAP), intuitively we asked whether lactylation would affect its activity due to the close proximity of the lactylated residue K147 to the C2 carbonyl group of FBP in the crystal structure of the enzyme/substrate complex[28] (Fig. 3a).

Previously, we site-specifically introduced lactylation into ALDOA in *E. coli*, and observed abolished enzyme activity with purified proteins[3]. Nevertheless, whether ALDOA-147Klac phenocopies this in vitro observation in living cells remains elusive. Since GCE allows producing ALDOA-147Klac in HEK293T cells, we are able to determine its effect on ALDOA enzyme activity. We first knocked down the endogenous ALDOA using small interfering RNA (siRNA) to eliminate the interference of endogenous ALDOA (the siALDOA group), followed by overexpression of ALDOA-WT (the siALDOA+WT group) or the genetically encoded ALDOA-147Klac (the siALDOA+147Klac group), respectively (Fig. 3b). We found that ALDOA knockdown significantly impaired enzyme activity when comparing the siALDOA group with cells transfected with scramble siRNA (the siCtrl group). Subsequent analysis of the siALDOA+WT group and the siALDOA+147Klac group showed that the impaired activity can be partially rescued by overexpression of ALDOA-WT, restoring the activity to approximately 85% of the siCtrl group, but not by overexpression of ALDOA-147Klac (Fig. 3c). Considering that the abundance level of overexpressed ALDOA was similar between the two groups (Fig. 3b), we confirmed that lactylation at K147 significantly inhibited ALDOA activity in vivo.

Given that ALDOA is essential for glycolysis, we hypothesized that dysfunctional ALDOA-147Klac may interfere with glycolysis. Metabolomic analysis of the siALDOA+WT and siALDOA+147Klac groups revealed that FBP—the glycolytic intermediate upstream of ALDOA—

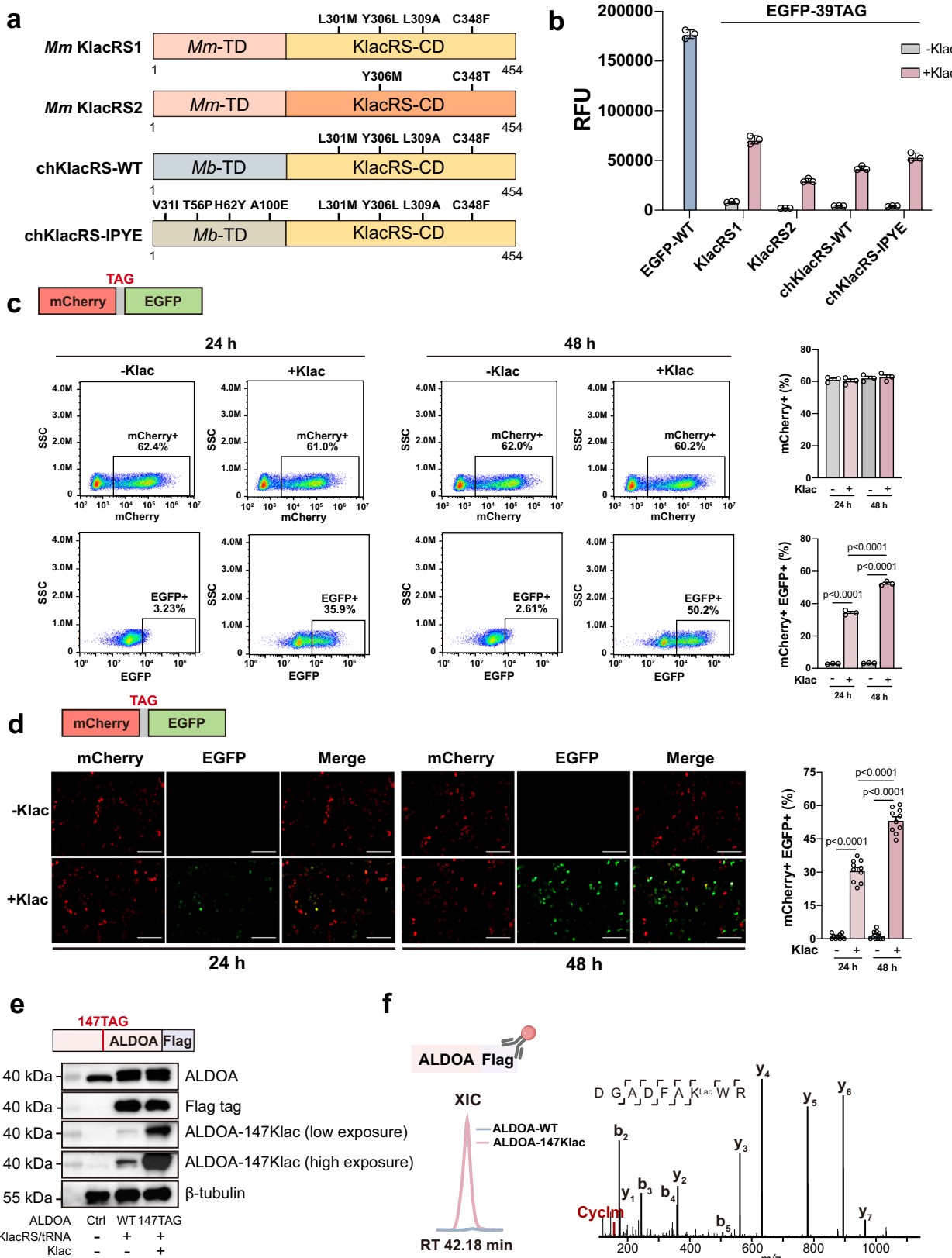

accumulated in the siALDOA+147Klac group compared to the siALDOA +WT group, whereas G3P and lactate (Lac)−the metabolites downstream of ALDOA−were reduced in the former group compared to the latter group (Fig. 3d). Consequently, the G3P/FBP and Lac/FBP ratios were reduced in siALDOA+147Klac group compared to the siALDOA +WT group (Supplementary Fig. 4), indicating inhibited ALDOA

activity and disrupted glycolytic flux in living HEK293T cells due to site-specific lactylation on ALDOA-K147. Seahorse analysis supported the observed changes−ALDOA knockdown decreased the extracellular acidification rate (ECAR) compared to normal cells, while ALDOA-WT overexpression largely restored the decreased ECAR. In contrast, ALDOA-147Klac failed to do so (Fig. 3e). These lines of evidence led us

**Fig. 2 | Introducing site-specific lactylation in living cells with genetic code expansion. a** Illustrated structures of KlacRS variants for site-specific incorporation of Klac. **b** Analysis of the incorporation efficiency of KlacRS variants by EGFP fluorescence assay. *E. coli* cells co-transformed with evolved or engineered KlacR-Ses/tRNA$_{CUA}^{Pyl}$ pairs and EGFP-39TAG plasmids in the presence or absence of Klac (1 mM, 16 h). Data represent the mean ± S.D. (*n* = 3 biological replicates/group). **c** Flow cytometry analysis of Klac incorporation efficiency in HEK293T cells co-transfected with the KlacRS1/tRNA$_{CUA}^{Pyl}$ pair and mCherry-TAG-EGFP plasmids in the presence or absence of Klac (1 mM, 24 h, and 48 h). Data represents the mean ± S.D. (*n* = 3 biological replicates/group) and the *p* value was calculated by one-way ANOVA. **d** Representative images of HEK293T cells co-transfected with the KlacRS1/tRNA$_{CUA}^{Pyl}$ pair and mCherry-TAG-EGFP plasmids in the presence or absence of Klac (1 mM, 24 h, and 48 h). Scale bar, 100 μm. Data represent the mean ± S.D. (*n* = 10

biological replicates/group), and the *p* value was calculated by one-way ANOVA. **e** Immunoblotting analysis of HEK293T cells expressing endogenous ALDOA, or overexpressing Flag-tagged ALDOA-WT or ALDOA-147Klac under the indicated transfection conditions, detected against anti-ALDOA, anti-Flag and the specific anti-ALDOA-147Klac antibodies. Low exposure: 2 s; high exposure, 6 s. The experiments were repeated three times with similar results. **f** Validation of successful incorporation of Klac on K147 of ALDOA. HEK293T cells overexpressing Flag-tagged ALDOA-WT or ALDOA-147Klac, followed by immunoprecipitation using anti-Flag antibody and bottom-up proteomic analysis. Left, extracted ion chromatogram (XIC) of the 147Klac-bearing peptide. The illustration was created in BioRender. Shao, C. (2024) BioRender.com/q85y323. Right, representative MS/MS spectrum with the signature CycIm ion revealing lactylation at K147. Source data are provided as a Source Data file.

to conclude that ALDOA-147Klac inhibits enzymatic activity, and thus suppresses glycolysis in living cells.

## Site-specific lactylation modulated ALDOA stability

Previous studies have demonstrated that PTM can alter not only the enzymatic activity but also the biophysical properties of modified proteins[29]. One extensively studied property is protein thermal stability (PTS)[30,31]. Changes in PTS can disrupt proteostasis and even cellular physiology[32]. The relationship between a specific PTM state and PTS has been inferred by comparing the melting curve of the modified proteoform with that of its unmodified counterpart[30]. Research has shown that site-specific phosphorylation on a serine residue located in the substrate binding domain of glyceraldehyde-3-phosphate dehydrogenase (GAPDH) can significantly destabilize the protein[30], whereas in certain cases O-linked N-acetylglucosamine (O-GlcNAc) can stabilize the modified proteins[31]. To investigate whether 147Klac of ALDOA may alter its thermostability, we subjected the living cells encoding ALDOA-WT and ALDOA-147Klac to a heating treatment according to the Cellular Thermal Shift Assay (CETSA) workflow[33]. We found that lactylation has a thermostabilizing effect on ALDOA based on the melting curves obtained by immunoblotting against the Flag tag (Supplementary Fig. 5a).

As the resistance to proteolysis is another indicator of protein stability, we also assessed the change in proteolytic stability of ALDOA-147Klac compared to ALDOA-WT using the drug affinity responsive target stability (DARTS)[34] workflow. We treated cell lysates with a proteolytic enzyme and performed immunoblotting against the Flag tag. We observed that the lactylated proteoform exhibited significantly greater resistance to proteolysis than unmodified ALDOA-WT (Supplementary Fig. 5b). This finding supports the lactylation-dependent stabilization detected by CETSA. Taken together, we anticipate that encoding lactylation in living cells via GCE may provide insight into the regulatory role of lactylation in protein stability and proteostasis, as well as exploring its implications in diseases[35].

## Lactylation altered subcellular distribution of ALDOA

After revealing lactylation on ALDOA-K147 as a loss-of-function PTM on enzyme activity, we sought to explore the gain-of-function capabilities of this site-specific modification. As PTMs provide a common and dynamic method to modulate the subcellular localization of modified proteins, as exemplified by metabolic enzymes such as GAPDH[36,37] and pyruvate kinase M2 (PKM2)[10,38], we asked that whether the lactylation on ALDOA-K147 can alter the partitioning behavior of ALDOA. To test this hypothesis, the plasmids mCherry-T2A-ALDOA (147TAG)-EGFP and mCherry-T2A-ALDOA (WT)-EGFP were designed. In these plasmids, ALDOA is fused to EGFP to visualize and track its distribution, while the T2A element allows simultaneous and separate expression of ALDOA-EGFP and mCherry, which serves as an internal transfection control to identify successfully transfected cells for subsequent analysis of ALDOA-EGFP from the mCherry$^+$ cells. The use of such plasmids allowed for real-time tracking and subcellular

quantification of ALDOA-WT/147Klac-EGFP in living cells using confocal microscopy.

First, we found that ALDOA-WT exhibited a clear cytoplasmic distribution by following the fluorescence signals of ALDOA-EGFP (Fig. 4a). This is consistent with its predicted localization by YLoc analysis[39] (Supplementary Fig. 6a) and previous reports of primary cytoplasmic localization of ALDOA in A-431, U-251, and U2OS cells (retrieved from the Human Protein Atlas[40], Supplementary Fig. 6b). Interestingly, in contrast to ALDOA-WT, fluorescence images showed that EGFP-tagged ALDOA-147Klac co-localized with Hoechst, indicating that ALDOA is distributed in both the nucleus and cytoplasm after lactylation (Fig. 4a). Statistical analysis confirmed that lactylation promoted the subcellular translocation of ALDOA to the nucleus (Fig. 4b).

To confirm the nuclear accumulation of ALDOA-147Klac, we performed subcellular fractionation experiments. HEK293T cells expressing His$_6$-tagged ALDOA-WT or ALDOA-147Klac were separated into cytoplasmic and nuclear fractions, and then were immunoblotted against His$_6$-tag and lactylation. The results showed that ALDOA-WT was primarily located in the cytoplasm, whereas ALDOA-147Klac enhanced its translocation to the nucleus (Fig. 4c), consistent with the microscopic observations (Fig. 4a).

Using the ALDOA-147Klac-specific antibody, we can directly assess the subcellular distribution of endogenous ALDOA-147Klac by immunofluorescence imaging. We found that ALDOA-147Klac was indeed enriched in the nucleus, whereas the unmodified form, ALDOA-WT, remained predominantly in the cytoplasm. When intracellular lactate levels were lowered following oxamate or FX-11 treatments, we confirmed that ALDOA-147Klac, especially the fraction localized to the nucleus, was significantly reduced (Fig. 4d, e). Since lactylation can modify different lysine residues on the same POI and this may induce distinct effects on subcellular localization, the GCE approach provides a valuable tool for elucidating the relationship between different lactylation states and protein localization (Fig. 4f).

## Lactylation on ALDOA induced transcriptional changes in living cells

Next, we reasoned that—as lactylation induced ALDOA to translocate into the nucleus—this might result in altered gene expression, based on previous findings on several metabolic enzymes, exemplified by PKM2[38], pyruvate decarboxylase (PDC)[41] and methylenetetrahydrofolate dehydrogenase 1 (MTHFD1)[42]. Driven by this speculation, we performed RNA-sequencing (RNA-seq) analysis on HEK293T cells site-specifically encoding ALDOA-147Klac by co-transfection of the KlacRS1/tRNA$_{CUA}^{Pyl}$ pair and ALDOA-147TAG plasmids and cultured with Klac (+Klac group), and the same transfected cells but cultured without Klac as a control group (-Klac group) (Supplementary Fig. 7a). Pairwise comparisons led us to identify 111 genes as differentially regulated (70 upregulated and 41 downregulated) by ALDOA-147Klac expression (Fig. 5a). Gene ontology (GO) analysis indicates that these genes are associated with various Biological Processes (BPs) and

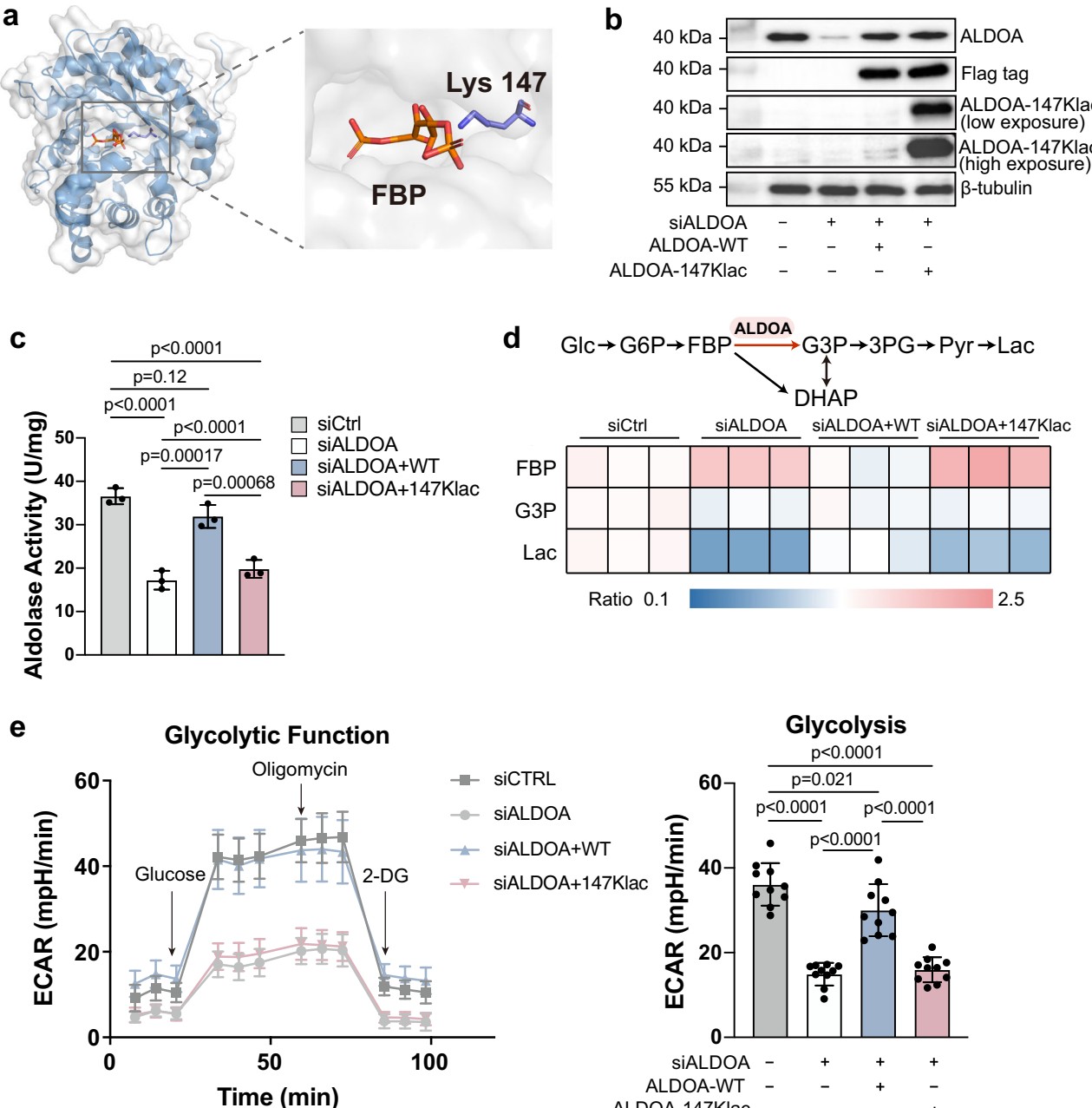

**Fig. 3 | Lactylation on ALDOA-K147 abolished enzyme activity and regulated glycolytic flux. a** Crystal structure of K147 in ALDOA (PDB 4ALD) and its substrate FBP. **b** Immunoblots of HEK293T cells expressing ALDOA-WT or ALDOA-147Klac after knocking down the endogenous ALDOA. **c** ALDOA activity of HEK293T cells expressing ALDOA-WT and ALDOA-147Klac after knocking down the endogenous ALDOA. Data represent the mean ± S.D. ($n$ = 3 biological replicates/group), and the $p$ value was calculated by one-way ANOVA. **d** Heat map comparing the abundance of metabolites involved in glycolysis. Abundance ratios were calculated by comparing ion intensities of individual metabolite in siALDOA, siALDOA+WT and siALDOA+147Klac groups, using the siCtrl group as a control. Glc glucose, G6P

glucose 6-phosphate, FBP fructose 1,6-bisphosphate, G3P glycerol-3-phosphate, DHAP dihydroxyacetone phosphate, 3PG 3-phosphoglycerate, Pyr pyruvate, Lac lactate. **e** Seahorse analysis of HEK293T cells expressing ALDOA-WT or ALDOA-147Klac after knocking down the endogenous ALDOA. The ECAR was measured in real-time under basal conditions and after the addition of glucose (10 mM), oligomycin (3 μM) and 2-DG (50 mM). Left, the time course of a representative experiment. Right, determination of glycolysis rate. Data represent the mean ± S.D. ($n$ = 10 biological replicates/group) and the $p$ value was calculated by one-way ANOVA. Source data are provided as a Source Data file.

Molecular Functions (MFs) (Fig. 5b and Supplementary Fig 7b). In particular, most of the genes were enriched in the BP of regulation of cell junction assembly. In accordance with this, KEGG pathway enrichment analysis revealed the enrichment of genes in the cell adhesion pathway (Supplementary Fig. 7c). Furthermore, Cellular Component (CC) analysis showed that most of the regulated genes were enriched in the basolateral plasma membrane (Supplementary Fig. 7d).

As these analyses appeared to suggest a plausible link between ALDOA-147Klac and cell adhesion/cell junction, we aimed to validate the hypothesis by focusing on the altered genes classified under the BP of cell junction assembly (Fig. 5a, b). Using RT-qPCR analysis, we confirmed the transcript-level changes detected by RNA-seq, including the downregulation of pro-adhesion genes, such as *CLDN1*[43], *LRRTM2*[44,45] and *SLITRK6*[45], and the upregulation of an anti-adhesion gene *GPBAR1*[46] (Fig. 5c and Supplementary Fig. 7e). Notably, the

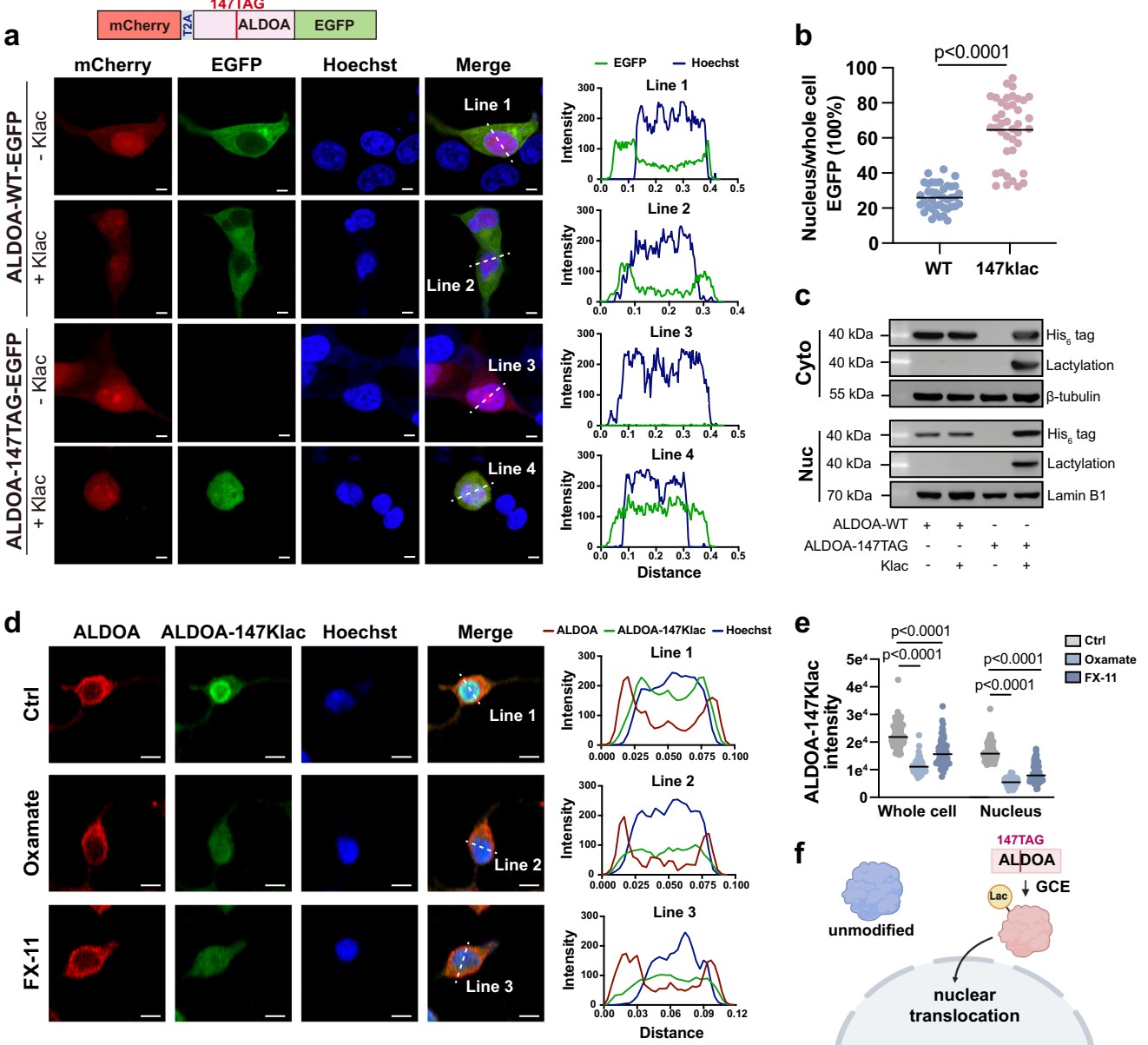

**Fig. 4 | Lactylation altered subcellular partition of ALDOA. a** Subcellular localization of EGFP-tagged ALDOA-WT or ALDOA-147Klac (green) analyzed by confocal microscopy in cells co-transfected with the KlacRS1/tRNA$_{CUA}^{Pyl}$ pair and the mCherry-T2A-ALDOA (WT/147TAG)-EGFP plasmids ($n = 40$ biological replicates/group). The nucleus was stained with Hoechst (blue), and mCherry (red) served as the transfection control. Left, representative images showing ALDOA-147Klac partially translocated into the nucleus compared to ALDOA-WT. Right, fluorescence intensity profiles across the indicated lines in the left. Scale bar, 5 µm. **b** Quantification of the percentage of nuclear ALDOA in the samples shown in (**a**), determined by normalizing the mean fluorescence intensity of nuclear EGFP-tagged ALDOA to that of the total EGFP-tagged ALDOA in whole cells. $n = 40$ biological replicates/group, and the $p$ value was calculated by unpaired two-tailed Student's $t$ test. **c** Subcellular localization of ALDOA-WT and ALDOA-147Klac determined by immunoblotting the cells co-transfected with the KlacRS1/tRNA$_{CUA}^{Pyl}$ pair and ALDOA-147TAG/ALDOA-WT plasmids in the presence or absence of Klac

(5 mM, 48 h). Cytoplasmic β-tubulin and nuclear lamin B1 were used as loading controls. Cyto, the cytoplasmic fraction; Nuc, the nuclear fraction. The experiments were repeated three times with similar results. **d** Representive immuno-fluorescence imaging of ALDOA and ALDOA-147Klac analyzed by confocal microscopy using cells treated without and with oxamate (25 mM, 24 h) or FX-11 (10 µM, 24 h). Left, HEK293T cells co-stained with the anti-ALDOA (red) and anti-ALDOA-147Klac antibodies (green) and Hoechst (Blue) ($n = 69$ for the control group, $n = 68$ for the oxamate-treated group and $n = 65$ for the FX-11-treated group). Right, fluorescence intensity profiles across the indicated lines in the left. Scale bar, 20 µm. **e** Analysis of the mean fluorescence intensity of whole-cell and nuclear ALDOA-147Klac for cells in (**b**). ($n = 69$ for the control group, $n = 68$ for the oxamate-treated group and $n = 65$ for the FX-11-treated group). The $p$ value was calculated by one-way ANOVA. **f** Illustration of site-specific lactylation at ALDOA-K147 inducing subcellular translocation. Created in BioRender. Shao, C. (2024) BioRender.com/r58q841. Source data are provided as a Source Data file.

enzyme activity of ALDOA was comparable between these two groups (+/-Klac groups), indicating that the transcriptional changes were caused by site-specific lactylation of ALDOA rather than altered ALDOA activity (Supplementary Fig. 7f).

To provide more solid evidence for the link between ALDOA-147Klac and adhesion-related transcript changes, we used two LDHA inhibitors, oxamate and FX-11, to reduce the ALDOA-147Klac levels without affecting enzyme activity (Supplementary Fig. 7 g). Upregulation of pro-adhesion genes, including *CLDN1*, *LRRTM2* and *SLITRK6*, and the downregulation of an anti-adhesion gene *GPBAR1*, were observed, in contrast to cells overexpressing ALDOA-147Klac (Fig. 5d).

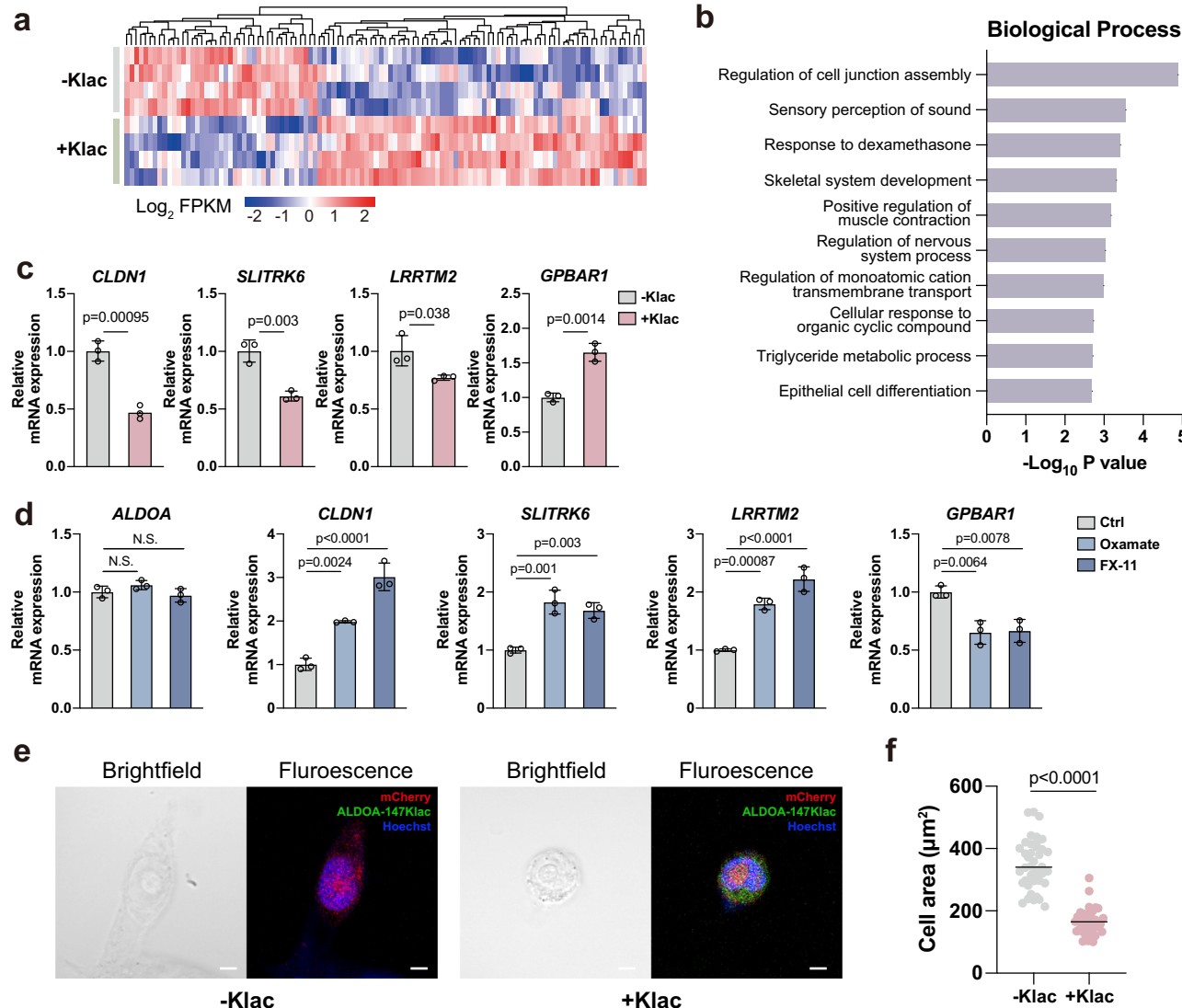

**Fig. 5 | Lactylation on ALDOA induced transcriptional changes in living cells.**
**a** RNA-seq analysis of cells co-transfected with the KlacRS1/tRNA$_{CUA}^{Pyl}$ pair and ALDOA-147TAG plasmids in the presence or absence of Klac (5 mM, 48 h). Heat map of genes showing significant changes (FC > 1.5 or < 0.67 and $p$ value < 0.05 by two-sided Wald test as implemented in DESeq2) by plotting the Log$_2$ FPKM value ($n$ = 40 biological replicates/group). FPKM, fragments per kilobase of transcript per million mapped fragments. **b** Bar plot of the GO BP analysis of the differentially regulated genes in (**a**) from Metascape. The $p$ values were calculated using one-sided Fisher's exact test and adjusted by the Benjamini-Hochberg method. **c** RT-qPCR analysis of *CLDN1*, *SLITRK6*, *LRRTM2* and *GPBAR1* expression levels using cells in (**a**). The endogenous β-tubulin gene was used as the internal control for normalizing the target gene levels. Data represent the mean ± S.D. ($n$ = 3 biological replicates/ group) and the $p$ value was calculated by one-way ANOVA. **d** RT-qPCR analysis of *ALDOA*, *CLDN1*, *SLITRK6*, *LRRTM2*, and *GPBAR1* expression levels in HEK293T cells treated with oxamate (25 mM, 24 h) or FX-11 (10 μM, 24 h). Data represent the mean ± S.D. ($n$ = 3 biological replicates/group) and the $p$ value was calculated by one-way ANOVA. **e** Representative brightfield and immunofluorescence images of HEK293T cells overexpressing (the +Klac group) or not overexpressing (the -Klac group) the EGFP-tagged ALDOA-147Klac (green) due to the availability of Klac (5 mM, 48 h) ($n$ = 40 biological replicates/group). The nucleus was stained with Hoechst (blue) and mCherry (red) serves as the transfection control. Scale bar, 5 μm. **f** Adhesive cell areas of cells in (**e**). $n$ = 40 biological replicates/group, and the $p$ value was calculated by unpaired two-tailed Student's t-test. Source data are provided as a Source Data file.

Changes in the expression levels of cell adhesion-related genes can lead to substantial morphological alterations[47]. Therefore, we characterized the cell morphology and adhesive area of cells with or without ALDOA-147Klac overexpression by culturing them in the presence or absence of Klac (+/-Klac). Microscopic observation revealed that cells expressing ALDOA-147Klac (the +Klac group) showed reduced size, a rounded appearance, and detachment from the substrate. In contrast, cells transfected with the same plasmids but without Klac administration (the -Klac group) displayed a more spread and adherent morphology (Fig. 5e). Accordingly, the adherent area of cells decreased significantly in the presence of Klac (Fig. 5f). Collectively, our findings suggest that ALDOA-147Klac regulated the expression of adhesion-related genes and influenced cell-matrix adhesion.

**ALDOA recruited different interacting partners after lactylation**
Proteins often localize to different subcellular niches to fulfil their discrete functions. Since lactylation of K147 altered the subcellular localization of ALDOA, we speculate that this modification may also change the recruitment of interacting proteins. To verify this, we performed co-immunoprecipitation (co-IP) experiments using ALDOA-WT-encoding and ALDOA-147Klac-encoding HEK293T cells (Fig. 6a) and sought to identify the changed interacting proteins of Flag-tagged ALDOA in response to lactylation with quantitative proteomics. Quantitative assessment of the co-immunoprecipitated proteins revealed marked differences between the ALDOA-147Klac interactome and the ALDOA-WT interactome (Fig. 6b, Supplementary Data 7). GO analysis revealed that the interactome of ALDOA-147Klac and ALDOA-

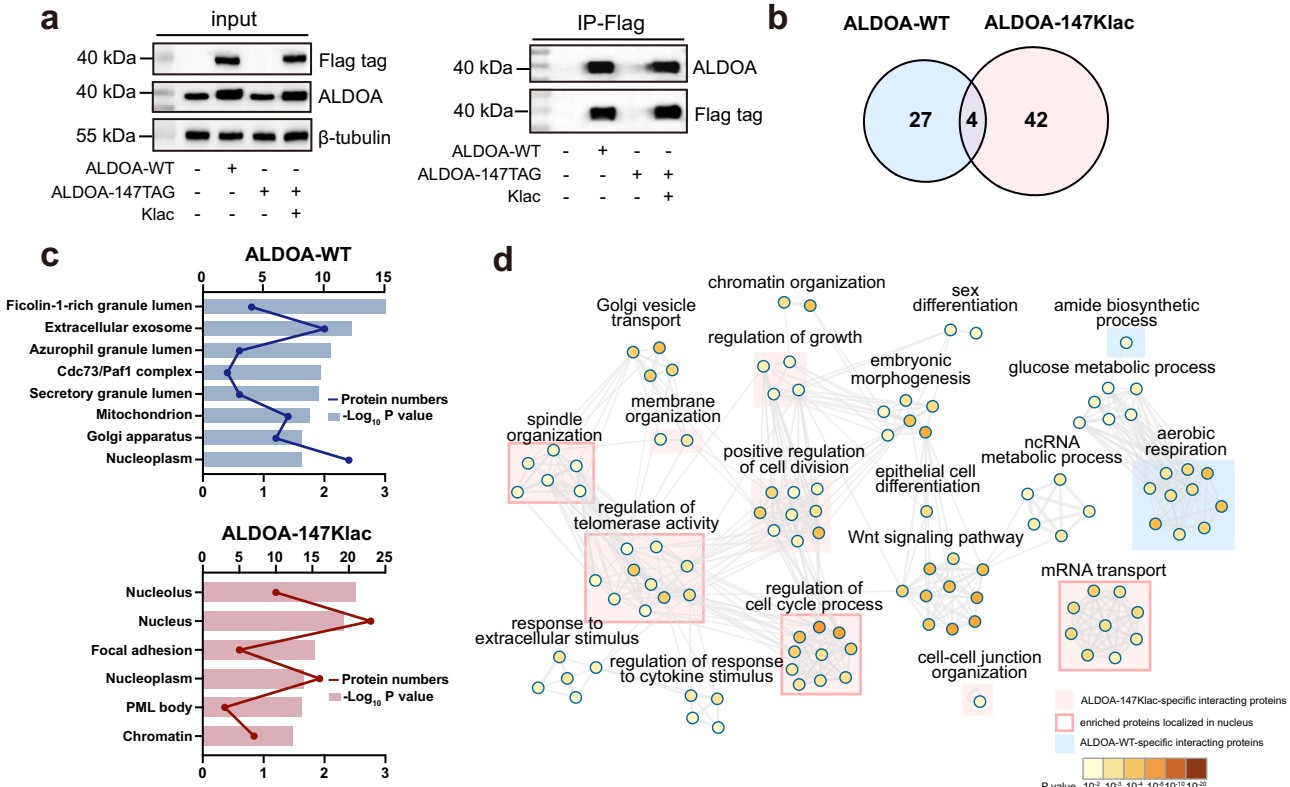

**Fig. 6 | ALDOA recruited different interacting partners after lactylation.**
**a** Immunoblotting confirmed the enrichment of Flag-tagged-ALDOA-WT/147Klac in HEK293T cells under the indicated transfection conditions. The experiments were repeated three times with similar results. **b** Venn diagrams showing the interacting proteins of ALDOA-WT and ALDOA-147Klac. The interacting proteins of ALDOA-WT were enriched by co-IP using anti-Flag magnetic beads using HEK293T cells transfected with the ALDOA-WT plasmids or vector, quantified by label-free quantification proteomics and screened with a cutoff of FC > 2 and $p$ value < 0.05 by unpaired two-tailed Student's $t$ test ($n = 3$ biological replicates/group). The interacting proteins of ALDOA-147Klac were enriched from HEK293T cells co-transfected with the KlacRS1/tRNA$^{Pyl}_{CUA}$ pair and ALDOA-147TAG in the presence or absence of Klac (5 mM, 48 h), quantified and screened using the same standard as ALDOA-WT ($n = 3$ biological replicates/group). **c** GO CC analysis of the interacting proteins in **b** from DAVID bioinformatics website. The $p$ values were calculated

using one-sided Fisher's exact test and adjusted by the Benjamini-Hochberg method. The lines indicate protein numbers enriched in each cellular component and the bars represent the −Log$_{10}$ $p$ value. **d** Network plot of the significantly enriched GO BPs for the interacting proteins of ALDOA-WT and ALDOA-147Klac in (**b**) using Metascape. Each node represents an enriched term and was colored by $p$ value using the one-sided Fisher's exact test and adjusted by the Benjamini-Hochberg method. The node size is proportional to the number of input genes falling into that term. GO terms with a similarity >0.3 are connected by edges, and the thickness of the edge represents the similarity score. The network was built with Cytoscape. The red shade indicates that >70% of the proteins enriched in the specific BP were identified only in the ALDOA-147Klac group. If >70% of the enriched proteins in the red-shaded BPs are located in the nucleus, the shades are further framed in red. Blue shade is used to similarly indicate proteins specifically identified in the ALDOA-WT group. Source data are provided as a Source Data file.

WT are linked to distinct sets of BPs, MFs and KEGG pathways (Supplementary Fig. 8a–c). CC analysis further led us to note the enrichment of the ALDOA-147Klac interactome in the nucleus (Fig. 6c)—this agreed with the lactylation-promoted nuclear translocation of ALDOA (Fig. 4). Accordingly, the interacting proteins of ALDOA-147Klac were selectively enriched in BPs primarily involving nuclear proteins. These BPs include regulation of cell cycle process, regulation of telomerase activity, mRNA transport and spindle organization (Fig. 6d). The interacting proteins of ALDOA-WT, which is primarily located in the cytoplasm, are enriched in metabolism-related BPs such as aerobic respiration and amide biosynthesis that also occur in the cytoplasm[48,49] (Fig. 6d). These findings indicate that distinct proteins were recruited to ALDOA as a consequence of lactylation.

## Discussion

Functional interrogation of lysine lactylation promises to identify therapeutic targets for lactate-related diseases. However, this endeavor has been hindered by the technologies available. Currently, the biological consequences of a specific lactylation on POIs are often pursued by mutating the lactylatable lysine residue to a non-lactylatable residue[2,4,6] and/or a lactylation-mimetic residue[5,7], and then studying the resulting biological changes. Nevertheless, such

mutagenesis cannot replicate the native structures and functions of lactylation. Alternatively, GCE allows for site-specific incorporation of PTM amino acids into POIs both in vitro and in vivo[8,50–56], and has therefore been used to study the functionality of acetylation[50], phosphorylation[51], sulfation[52,53], butyrylation[54] and lipidation[55]. In contrast, lactylation has rarely been genetically encoded by GCE in POIs other than model proteins[8], and thus its potential remains largely untapped. As the amount of functional lactylproteome data continues to grow, the need for an efficient GCE approach is increasing. Therefore, here we developed an efficient and orthogonal GCE system to genetically encode protein lactylation in living cells and to elucidate the functionality of lactylation on POIs.

First, using a wet-and-dry-lab combined strategy we discovered lactylation of biological importance through analyzing diverse proteome datasets. A particular lactylation, on ALDOA-K147, is present in all assayed human cell lines and most human tissues, as well as in lower non-human species such as rabbits and fruit flies. Considering that recent studies on lactylation functionality have mostly focused on histones and are therefore limited to transcriptional regulation[1,14–18], pursing the biological outcomes of non-histone lactylation is anticipated to significantly enrich and expand our understanding towards lactylation. After successful expression of ALDOA-147Klac in

mammalian cells via GCE, we found that this modification can inhibit enzyme activity, and impair glycolytic flux. This is consistent with our previous in vitro observation using purified ALDOA-147Klac protein, harvested from *E. coli*[3]. These lines of evidence demonstrate a causal relationship between lactate and glycolysis homeostasis—lactate can control its biosynthetic pathway by inhibiting its upstream enzyme ALDOA through lactylation. This is not surprising as acylation installed in the active site of metabolic enzyme also results in a loss-of-function in terms of enzymatic activity[10,38].

Intriguingly, our established IFA platform revealed that lactylation is also a gain-of-function modification. For instance, we discovered that ALDOA-147Klac altered the subcellular distribution of ALDOA and promoted its nuclear accumulation. Actually, in addition to lactylation, PTMs such as acetylation[10], succinylation[38], phosphorylation[57] and S-glutathionylation[36] also act as relocalization signals for metabolic enzymes. However, the mechanisms underlying nucleocytoplasmic translocation following lactylation are not fully understood. The GCE approach may facilitate the investigation of the dependent machinery. For example, we can selectively silence or inhibit the components hypothetically involved in this process, such as importins and exportins[58], and then monitor the subcellular partitioning of lactylated ALDOA, introduced by GCE, to infer the causal relationships.

As nuclear relocalization often enables POIs to participate in gene transcription[59,60], we intuitively examined transcriptome changes in cells following the overexpression of the nucleus-localized ALDOA-147Klac. We noted affected transcription for certain adhesion-related genes, and this change could further lead to cellular morphological alterations. This finding supports previously reported "moonlighting" roles of metabolic enzymes[42,57]—metabolic enzymes are discovered to have non-metabolic functions when redistributed from the cytosol or mitochondria to the nucleus, and can govern a wide range of cellular processes, including influencing histone PTM patterns, interacting with transcription factors, altering chromatin structure and mRNA stability. These diverse mechanisms all converge in altered gene transcription[60]. We propose that, by using the GCE approach, an accurate roadmap linking the subcellular localization of POIs, along with the resulting biological consequences such as gene transcription, to site-specific lactylation can be generated in the future.

In addition to the translocation machinery, it is also applicable to deciphering the regulatory machinery of lactylation. The candidate writer/eraser of lactylation can be conveniently identified by treating the lactylation-encoding cells with genetic or pharmacological interventions that selectively target the candidate writer/eraser enzymes and assessing the changes in lactylation levels. Proteomics methods for the identification of protein-protein interactions (PPIs) are promising to discover previously unreported writers, erasers and readers for lactylated POIs. Unfortunately, in this study, we were unable to identify the regulatory enzymes responsible for ALDOA-147Klac using the co-IP method, whereas photocrosslinking-based PPI approaches and proximity labeling may offer more effective strategies for capturing weak and transient PPIs between lactylated POIs and their associated writers, erasers and readers[61–63].

Limitations of the current GCE method of lactylation incorporation in living cells still exist and warrant future optimization. First, our method using the orthogonal KlacRS/tRNA$_{CUA}^{Pyl}$ pair, is based primarily on transient transfection[8,12,13,52,53,55]. This limits the scope of genetically encoded lactylation only to cell lines that allow efficient transfection. Future adaptation of the GCE method with PiggyBac transgenesis[50] or CRISPR-based genome editing[54] is anticipated to generate stable cell lines with efficient and uniform incorporation of lysine lactylation. Second, we employed bottom-up proteomics to uncover the lactylation sites naturally present in cells and direct the appropriate sites for Klac incorporation. Nevertheless, bottom-up analysis is competent in detecting site-specific PTMs through proteome-wide analyses, but cannot measure intact proteoforms of POIs and hence suffers from the loss of PTM combinatorial information[64]. Continuous efforts will be made to use the complementary proteomics approaches to uncover how lactylation may be installed at distinct residues of POIs simultaneously, and work in combination to modulate protein functions. It is also possible that lactylation co-exist with other PTMs on the same POI[65]; the ability to reveal how these PTMs crosstalk and act in a concerted manner to regulate protein functionality in living cells will deepen our understanding of lactylation from a comprehensive view of the PTM landscape than studies focusing on lactylation alone. Therefore, future efforts towards developing GCE methods that enable simultaneous incorporation of lactylation and other PTMs in POIs are warranted.

In summary, we have developed a research workflow for lactylation that assembles proteomic mining, the GCE approach, and IFA. This workflow is amenable to study lactylation of any POIs in living cells. It equips us with the ability to accurately assess the perturbed activity, stability, spatial distribution, interacting proteins of POIs and cell-wide changes caused by site-specific lactylation. The developed workflow is expected to spur research in the field of lactylation by helping to unravel the enigmatic functionality of site-specific lactylation in cells and even organisms. Such knowledge will open up the possibility of investigating whether modulation of site-specific lactylation could be a therapeutic approach for diseases, particularly those associated with metabolic disorders.

## Methods

### Cell culture and chemicals

THP-1 and MC38 were purchased from Cell Bank/Stem Cell Bank, Chinese Academy of Sciences (Shanghai, China). HEK293T, HCT116, PC3, H1299, HepG2, A549, TALL-104, RAW264.7, BV2 and LLC cells were purchased from American Type Culture Collection (ATCC) and cultured at 37 °C in a 5% $CO_2$ atmosphere. HEK293T, HepG2, RAW264.7, BV2 and LLC were cultured in Dulbecco's Modified Eagle Medium (DMEM). HCT116, PC3, A549, H1299, THP-1 and MC38 were cultured in RPMI-1640 medium. TALL-104 was cultured in RPMI-1640 medium containing 10 ng/mL interleukin-2 (PeproTech, cat. no. 200-02). All culture media were purchased from Gibco and supplemented with 10% fetal bovine serum (Excell, cat. no. FSP500), 100 U/mL penicillin and 1 μg/mL streptomycin (Thermo Fisher Scientific, cat. no. 15070-063). LC-MS grade water and acetonitrile (ACN) were obtained from Merck (Darmstadt, Germany). All chemicals were purchased from Sigma-Aldrich unless otherwise specified. The oligonucleotide primers were obtained from Tsingke (Nanjing, China).

### Dry-lab strategy for lactylation mining

For lactylation mining based on public proteome data, database searching parameters were set according to the mass spectrometer and parameters used in the literature. Specifically, Meltome Atlas data[23] were searched using PEAKS Studio XPro (Bioinformatics Solutions Inc.) against the UniProt human proteome database (UniProt_Human_reviewed_29-11-2021.fasta). A mass tolerance of 5 ppm was allowed for precursor ions and 0.02 Da for fragment ions. Human tissues proteome data[24] were searched using PEAKS Studio online 11 against the UniProt human proteome database (UniProt_Human_reviewed_20-7-2023.fasta). A mass tolerance of 10 ppm was allowed for precursor ions and 0.05 Da for fragment ions. Proteome data collected from different animal species were searched using PEAKS Studio Xpro against their species-specific UniProt FASTA database. Specifically, UniProt_Human_reviewed_15-5-2023.fasta for *Homo sapiens*, UniProt_Mouse_reviewed_11-6-2023.fasta for *Mus musculus*, UniProt_Rabbit_reviewed_26-6-2023.fasta for *Oryctolagus cuniculus*, UniProt_Fruit fly_reviewed_27-6-2023.fasta for *Drosophila melanogaster* and UniProt_Western flower thrips_unreviewed_7-12-2023.fasta for *Frankliniella occidentalis* were used. A mass tolerance of 7 ppm was allowed for precursor ions and 0.02 Da for fragment ions. For other parameters, we set trypsin as the protease and allowed a

 

maximum of two missed cleavages and semi-specific digestion. Carbamidomethylation of cysteine (+57.02 Da) was set as fixed modification and methionine oxidation (+15.99 Da), lysine lactylation (+72.02 Da) and acetylation at protein N termini (+42.02 Da) were set as variable modifications. Peptide-spectrum matches (PSMs) were filtered to 1% FDR employing a target-decoy database search approach. The modified peptides reached an AScore>20[66] was assigned as lactylated candidates. Further, MS/MS spectra of such peptides were manually confirmed.

## Wet-lab strategy for lactylation mining

**Sample preparation.** To induce lactylation, cells were treated with 25 mM sodium lactate for 24 h, washed with cold PBS three times, harvested and lysed in RIPA lysis buffer (Beyotime, cat. no. P1003B) with protease and phosphatase inhibitor cocktail (ApexBio, cat. no. K1007 and K1013). Then, methanol, chloroform and water were added to the lysate at a ratio of 4:1:3:1 by volume. Precipitated proteins were collected by centrifugation at 12,000 rpm for 10 min and washed twice with methanol, followed by re-solubilized by 8 M urea in 25 mM ammonium bicarbonate solution. Proteins were then reduced by 10 mM dithiothreitol (DTT) at 56 °C for 30 min and alkylated by 40 mM iodoacetamide (IAM) at 25 °C for 20 min in dark. Additional DTT was added to react with excess IAM at 25 °C for 10 min. Subsequently, the mixtures were added with 25 mM ammonium bicarbonate to dilute urea to 1 M, followed by digestion with sequencing-grade trypsin (Promega, cat. no. V5111) at an enzyme/protein ratio of 1:50 (w/w) overnight at 37 °C. The lactylated peptides were enriched by immunoprecipitation according to literature[3]. Briefly, protein A agarose beads (Invitrogen, cat. no. 15918014) were washed with ETN buffer (1 mM EDTA, 50 mM Tris-HCl (pH 8.0) and 100 mM NaCl) three times, blocked with 5% BSA and conjugated with anti-lactylation antibody at 4 °C overnight. Then, the digested peptides were resolved with ETN buffer and incubated with the prepared antibody-conjugated beads with rotary shaking at 4 °C for 6 h. The beads were washed three times with ETN buffer, and the beads-bounded peptides were eluted with 1% trifluoroacetic acid in 40% ACN. The eluted peptides were desalted with $C_{18}$ Zip-tips, evaporated to dryness and stored at -80 °C prior to analysis.

**Lactylproteome analysis.** The digested cell lysates were analyzed by an Orbitrap Eclipse Tribrid mass spectrometer equipped with an EASY-nano LC 1200 system (Thermo Fisher Scientific). The mobile phase consisted of solvent A (0.1% FA in water) and solvent B (ACN/Water, 8:2, v/v). The flow rate was set at 300 nL/min. Peptides were analyzed using an Acclaim PepMap RSLC column (75 μm × 250 mm, Thermo Fisher Scientific) with a 90-min chromatography gradient: 0-5 min, 3-8% phase B; 5-60 min, 8-28% phase B; 60-75 min, 28-38% phase B; 75-80 min, 38-100% B; 80-90 min, 100% B. For MS data acquisition, MS1 spectra were collected at the $m/z$ range of 350–1800 at a resolution of 120,000 on the Orbitrap with a maximum AGC value of $4e^5$. For MS2 acquisition, fragmentation was conducted by HCD with the CE value set at 32% after optimization. MS2 spectra were collected with the first mass set at $m/z$ 110 at a resolution of 30,000 on the Orbitrap with a maximum AGC of $5e^4$.

**Data analysis.** Lactylproteome data were searched using similar parameters as dry-lab analysis with PEAKS Studio XPro against the UniProt human proteome database (UniProt_Human_reviewed_15-5-2023.fasta) and UniProt_Mouse_reviewed_11-6-2023.fasta. Specifically, mass tolerance was set to 10 ppm for precursors and 0.02 Da for fragment ions, and FDR was filtered to 1% for PSMs. No additional CycIm ion filtering is required.

## Phylogenetic analysis

Protein sequences of ALDOA from each species were aligned with the Molecular Evolutionary Genetics Analysis X (MEGA X) software[67]. The phylogenetic tree was constructed by MEGA X using the Neighbor-Joining method with 1000 bootstrap replicates. The tree is drawn to scale, with branch lengths in the same units as those of the evolutionary distances used to infer the phylogenetic tree. The evolutionary distances were computed using the Poisson correction method. The scale bar indicates the number of amino acid substitutions per site.

## KlacRS selection and engineering

The focused mutant library of pBK-*Mm*PylRS was constructed and transformed into DH10B cells containing pREP positive selection plasmid via electroporation. The cells were recovered with 1 mL pre-warmed SOC medium and shaken vigorously at 37 °C for 1 h. The recovered cells were plated on LB agar plate containing 50 μg/mL kanamycin (Kan), 25 μg/mL tetracycline (Tet), 100 μg/mL chloramphenicol (Cm), 0.02% L-arabinose (Ara) and 1 mM Klac. The plate was cultivated at 37 °C for 48 h. Colonies with strong green fluorescence were picked and replicated on LB agar plates containing 50 μg/mL Kan, 25 μg/mL Tet, 100 μg/mL Cm and 0.2% Ara with or without 1 mM Klac. A clone exhibiting Klac-dependent growth and fluorescence was named as pBK-*Mm*KlacRS2 and Sanger sequenced. To compare its incorporation efficiency with other KlacRS variants, *Mm*KlacRS2 was then cloned into pEvol vector by homologous recombination following the manufacturer's instructions (Vazyme, cat. no. C112).

To construct the chimeric KlacRS clones, namely pEvol-chKlacRS-WT and pEvol-chKlacRS-IPYE, we fused residues 1–149 of either wild type *Mb*PylRS (*Mb*PylRS-WT) or its activated mutant (*Mb*PylRS-IPYE) with residues 185-454 of *Mm*KlacRS1 by overlapping PCR and then cloned into pEvol vectors by homologous recombination. The DNA sequences of KlacRS variants are shown in Supplementary Data 5 and the primers used are listed in Supplementary Data 6.

## Incorporation efficiency of KlacRS variants

The plasmids pEvol-*Mm*KlacRS1, pEvol-*Mm*KlacRS2, pEvol-chKlacRS-WT, and pEvol-chKlacRS-IPYE were co-transformed into DH10B cells with pBad-EGFP-39TAG reporter, respectively. The transformants were grown in 2 mL LB containing 100 μg/mL ampicillin (Amp) and 50 μg/mL Cm at 37 °C until OD600 reached 0.6, followed by adding 0.2% Ara with or without 1 mM Klac. After being grown at 30 °C for another 16 h, cells were collected by centrifugation, washed once with PBS, and then resuspended in PBS. The relative fluorescence units (RFU) of cells were determined by dividing the absolute fluorescence by the OD600 readings of each sample using BioTek Synergy H1 microplate reader (Agilent, Santa Clara, CA, USA) to compare the incorporation efficiency of above KlacRS variants. The wild-type EGFP cultured under the same conditions was used as a control.

## Cell transfection, Klac incorporation and fluorescence imaging

For siRNA transfection, scramble siRNA (siCtrl) and ALDOA siRNA (siALDOA) were purchased from Corues Biotechnology (Nanjing, China). The sequence for siCtrl is 5′-UUCUCCGAACGUGUCACGUTT-3′ and for siALDOA is 5′-CCGAGAACACCGAGGAGAATT-3′. Approximately $3 × 10^5$ HEK293T cells were seeded into 6-well plates and transfected with 15 nM siRNA using 5 μL lipofectamine™ RNAiMAX reagent (Thermo Fisher Scientific, cat. no. 13778150) for 48 h according to the manufacturer's instructions. The efficiency of silencing was confirmed by immunoblotting.

To genetically encode Klac into target proteins, ~$3 × 10^5$ HEK293T cells were seeded into 6-well plates and cultured for 24 h. Plasmids pCMV-EGFP-Y39TAG and pcDNA-mCherry-TAG-EGFP were co-transfected with the pNEU-*Mm*KlacRS1 plasmid, respectively, using PolyJet transfection reagents according to the manufacturer's instructions. Then, the culture media were replaced with fresh complete medium with or without indicated concentration of Klac at 18 h post-transfection and cells were cultured for another 24 or 48 h. The transfected cells were examined by fluorescence imaging using Leica

DMI 3000B light microscope (Leica, Wetzlar, Germany) and relative fluorescence intensities between cells cultured with or without Klac were calculated by ImageJ (National Institutes of Health, V1.51). Following the same procedures, pCMV-ALDOA-147TAG or pCMV-ALDOA-WT plasmids were co-transfected with pNEU-*Mm*KlacRS1 and subjected to immunoblotting analysis.

## Flow cytometry analysis

Approximately $3 \times 10^5$ HEK293T cells were seeded into six-well plates and cultured for 24 h. Plasmids pcDNA-mCherry-TAG-EGFP or pCMV-EGFP-Y39TAG were co-transfected with pNEU-*Mm*KlacRS1 and cultured in the presence or absence of the indicated concentration of Klac for additional 24 h and 48 h. Cells were washed with cold PBS, harvested by centrifugation and resuspended in PBS. The fluorescence of cells was measured by CytoFlex flow cytometer using 405/488 nm lasers (Beckmann Coulter, CA, USA).

## Immunoblotting

Cells were lysed in RIPA lysis buffer supplemented with protease and phosphatase inhibitor cocktail. Nuclear and Cytoplasmic Protein Extraction Kits (Beyotime, cat. no. P0027) were used for separating cytoplasmic and nuclear proteins. Protein concentrations were first determined by the bicinchoninic acid (BCA) assay (Beyotime, cat. no. P0011). Then, the lysates were diluted by 4×XT Sample Buffer (Bio-rad, cat. no.1610791), heated to 100 °C for 5 min, cooled and separated by 10% SDS-PAGE gels. Proteins were then transferred onto polyvinylidene difluoride (PVDF) membranes (Bio-rad, cat. no.1620177), blocked with 5% non-fat dry milk in Tris-buffered saline with 0.1% Tween-20 detergent (TBST) and incubated with primary antibodies at 4 °C overnight. The membranes were subsequently washed five times with TBST and incubated with horseradish peroxidase (HRP)-conjugated secondary antibody for 1 h at 37 °C. The immunoblotted bands were detected by the addition of HRP substrate (Bio-rad, cat. no. 1705601), captured on a ChemiDoc XRS+ system (Bio-rad, Hercules, USA) and analyzed by ImageLab software. The primary antibodies used in this study include the antibody against ALDOA (Proteintech, 11217-1-AP, 1:1000), His₆-tag (Cell Signaling Technology, 2365S, 1:1000), Flag tag (Cell Signaling Technology, 14793S, 1:1000), ALDOA-147Klac (PTM Bio Inc, customized, 1:1000), lysine lactylation (PTM Bio Inc, PTM-1401RM, 1:2000), β-tubulin (Proteintech, 10068-1-AP, 1:1000) and β-actin (Proteintech, 66009-1-Ig, 1:20000).

## ALDOA activity assay

Enzymatic activity of ALDOA-WT and ALDOA-147Klac were assessed using an aldolase activity colorimetric assay kit (Biovision, cat. no. K665-100) according to the manufacturer's instructions as previously reported[3]. Briefly, cells were collected and homogenized with ice-cold aldolase assay buffer and kept on ice for 10 min. Cell lysates were centrifuged, and the supernatant was collected. The supernatants were then diluted with the assay buffer to prepare the sample solution. Different volumes of NADH standard were used to generate a NADH standard curve. The reaction solution was prepared by mixing the aldolase enzyme mix, aldolase developer, aldolase substrate and aldolase assay buffer. The reaction was initiated by the addition of 50 μL reaction solution to 50 μL sample solution and standard solution, respectively. ALDOA activity was measured photometrically by monitoring the absorbance at 450 nm in kinetic mode for 30 min with a 1-min interval at 37 °C. Data were normalized to the protein concentrations as determined by BCA assay for each sample.

## Metabolomic analysis of glycolytic metabolites

Cells were washed twice with cold PBS and intracellular metabolites were extracted by the addition of 1 mL extraction solvent that consisted of methanol/ACN/water (40:40:20, v/v/v, pre-cooled at −20 °C) and 0.5 μg/mL 4-chloro-phenylalanine as an internal standard (IS). Following co-

incubation for 30 min at −20 °C, cells were scraped off and the mixtures were transferred to 1.5 mL centrifugation tubes, followed by centrifugation at 18,000 × *g* for 10 min at 4 °C. The resultant supernatant was collected and evaporated to dryness, then analyzed by LC/MS.

Metabolites were analyzed using an Agilent UHPLC 1290 (Agilent Technologies, Santa Clara, CA USA) coupled to an Agilent 6546 Q/TOF mass spectrometer. Chromatographic separation was achieved using an XBridge Amide HPLC column (100 × 4.6 mm, 3.5 μm, Waters). The mobile phase consisted of solvent A (10 mM ammonium acetate in water, pH adjusted to 9.0 by ammonia) and solvent B (ACN). The gradient was set as follows: 0-1 min, 85% B; 1-16 min, 85-30% B; 16-18 min, 30% B; 18-18.5 min, 30-85% B; 18.5-30 min, 85% B. The flow rate was set at 0.4 mL/min and the column temperature was set as 40 °C. Following separation, the Agilent 6546 Q-TOF was operated in the negative mode (ESI−) for metabolite detection. The following parameters were used: liquid nebulizer, 45 psi; nitrogen drying gas, 8 L/min; drying gas temperature, 320; ESI capillary voltage, -3500 V; MS scan, *m/z* 50-1100; fragmentor voltage, 80 V. The Agilent Masshunter Workstation Data Acquisition was used for data acquisition and Agilent Masshunter Qualitative Analysis for data analysis. Metabolite identification was performed by matching the retention time, the precursor *m/z* and the fragment ions against analytical standards. Data were normalized to the total ion abundance as determined by Progenesis QI (version 2.0, Waters).

## Metabolic measurement by seahorse

For Seahorse analysis, ECAR of cells was measured using the Seahorse XF Glycolysis Stress Test Kit (Agilent Technologies, Palo Alto, CA, USA) on an XFe96 extracellular flux analyzer (Seahorse Bioscience, Billerica, MA, USA). Briefly, $1 \times 10^5$ HEK293T cells were seeded onto Seahorse XF96 microplates and cultured overnight. Then, the cultured media were replaced with the assay medium (XF base medium containing 2 mM glutamine, pH = 7.4), and cells were equilibrated at 37 °C in a CO₂-free incubator for 1 h. After baseline measurement, the standard Glycolysis Stress test was carried out with the stepwise injection of 10 mM glucose, 3 μM oligomycin and 50 mM 2-deoxy-D-glucose (2-DG). Metabolic parameters were collected and analyzed using the XF Wave software (Agilent, Palo Alto, CA, USA). Data were normalized to the protein concentrations as determined by BCA assay for each well.

## DARTS assay

HEK293T cells expressing Flag-tagged ALDOA-WT and ALDOA-147Klac were lysed in M-PER buffer (Thermo Scientific, cat. no. 78501) supplemented with protease and phosphatase inhibitor cocktail. Protein concentrations were determined by BCA assay. Cell lysates were diluted with TNC buffer (50 mM Tris-HCl at pH 8.0, 50 mM NaCl and 10 mM CaCl₂) to 2 mg/mL and then divided into seven aliquots of 40 μL, respectively. Different concentrations of Pronase (Roche, cat. no. 10165921001) were added to aliquoted lysates followed by incubation at 25 °C for 30 min. Proteolysis was quenched by the addition of 4×XT Sample Buffer and boiling for 10 min. Samples were then subjected to immunoblotting analysis using antibodies against the Flag tag and β-actin. The resultant band intensities of the Flag-tagged ALDOA were normalized to the intensity of the Pronase-untreated group and fitted to the Boltzmann sigmoid equation using Prism v.8.0.1 (GraphPad software).

## CETSA

HEK293T cells expressing ALDOA-WT and ALDOA-147Klac were aliquoted, followed by heating at designed temperatures ranging from 56 to 77.5 °C for 3 min in the Thermal Cycler (2720 Thermal Cycler, Applied Biosystems, CA, USA) and cooling down at room temperature for another 3 min. All samples were then subjected to three freeze-thaw cycles and centrifuged at 20,000 × *g* for 20 min. The soluble fractions were diluted by 4×XT Sample Buffer, heated to 100 °C for 5 min, and subjected to immunoblotting analysis. The resultant band

intensities of the Flag-tagged ALDOA were normalized to the intensity of the unheated group and fitted to the Boltzmann sigmoid equation using Prism v.8.0.1 (GraphPad software)[68]. The melting temperature ($T_m$) is defined as the temperature at which a 50% reduction in signal (soluble protein) is observed.

## Immunofluorescence staining and confocal imaging
HEK293T cells expressing ALDOA-WT and ALDOA-147Klac were plated in 35 mm glass-bottom dishes (Nest, Jiangsu, China). Cells were washed with PBS, fixed in 4% paraformaldehyde for 15 min at room temperature in dark, and permeabilized with 0.1% Triton X-100 for 10 min. Cells were then incubated with Hoechst 33258 (KeyGEN, cat. no. KGA1802) for 5 min at room temperature in dark and washed with PBS. For the immunofluorescence assay, cells were seeded in 35 mm glass-bottom dishes for attachment, then washed with PBS and fixed in 4% paraformaldehyde for 15 min at room temperature in the dark. Following permeabilization with 0.1% Triton X-100, cells were blocked with 1% BSA for 1 h, then incubated overnight at 4 °C with primary antibodies against ALDOA (Santa Cruz, sc-377058, 1:200) and ALDOA-147Klac (PTM Bio Inc, customized, 1:200). After washing with PBST (0.01% Tween-20 in PBS), cells were incubated for 1 h at room temperature with CoraLite® Plus 488 goat anti-rabbit IgG (Proteintech, RGAR002, 1:400) and Alexa Fluor® 647 donkey anti-mouse IgG (Abcam, ab150107, 1:400). Following another wash with PBST, cells were stained with Hoechst 33342 (KeyGEN, cat. no. KGA1805) for 15 min at room temperature in the dark. Images of cells were acquired using a Zeiss 700 confocal microscope (Zeiss, Jena, Germany).

## RNA-seq analysis
Total RNA was extracted using TRIzol reagent (Thermo Scientific, cat. no. 15596026), and RNA-seq was performed by Annoroad Gene Technology (Beijing, China). Briefly, the integrity and concentration of RNA were analyzed using an Agilent RNA 6000 Nano Kit (Agilent, cat. no. 5067-1511) and Agilent 2100 Bioanalyzer (Agilent Technologies). RNA-seq was performed using an Illumina Nova 6000 platform (Illumina), and the Novaseq Control Software (v.1.7.5) was used for data collection. Raw sequenced reads were filtered to achieve high-quality reads and then mapped to the human genome (GRCh38.100.chr) using HISAT2. Differentially expressed genes (DEGs) were analyzed using the R package DESeq2 with a cutoff of FC > 1.5 or <0.67 and $P$ value < 0.05 ($n = 4$ biologically independent samples/group). DEGs were subjected to KEGG pathway analysis and GO functional annotation including molecular function, biological process and cellular component by Metascape[69] (https://metascape.org/). A setting of group $P$ value < 0.01 and the inclusion of at least three genes in each group was used for filtering.

## RNA extraction and RT-qPCR analysis
Cells were harvested and washed with PBS twice. Total RNA was isolated using FreeZol Reagent (Vazyme, cat. no. R711-01) according to the manufacturer's instructions. Approximately 200 ng of total RNA was reverse transcribed into cDNA using Hiscript II RT SuperMix (Vazyme, cat. no. R222-01). RT-qPCR analysis of target genes was then performed using SYBR qPCR Master Mix (Vazyme, cat. no. Q341-02) on a real-time PCR cycler (StepOne Plus, Applied Biosystems, CA, USA). The endogenous β-tubulin gene was used as a reference to normalize the expression levels of target genes. The primer sequences used in this study are listed in Supplementary Data 6.

## Co-IP analysis of ALDOA interactome
Cells were harvested and lysed in 0.5% NP-40 lysis buffer (Beyotime, cat. no. P0013F) with protease inhibitor cocktail, phosphatase inhibitor cocktail and super nuclease (Beyotime, cat. no. D7271) followed by centrifugation at 18,000 × $g$ for 10 min. Protein concentration was determined by BCA assay, and ~1 mg of protein lysates were incubated with anti-Flag magnetic beads (Selleck, cat. no. B26101) for 4 h at room temperature for immunoprecipitation. For immunoblotting, the beads-bound proteins were eluted with SDS loading buffer and then heated at 100 °C for 5 min. For the ALDOA interactome analysis, the beads-bound proteins were then washed with PBS three times and incubated with 8 M urea in 25 mM ammonium bicarbonate solution for denaturation. Proteins were reduced by 10 mM DTT at 56 °C for 30 min and alkylated by 40 mM IAM at 25 °C for 20 min in the dark. Additional DTT was added to react with excess IAM at 25 °C for 10 min. Subsequently, the mixtures were added with 25 mM ammonium bicarbonate for diluting urea to 1 M, followed by digestion with trypsin at an enzyme/protein ratio of 1:50 (w/w) overnight at 37 °C. Samples were then acidified, desalted with $C_{18}$ Zip-tips, evaporated to dryness, followed by analyzed using the Orbitrap Eclipse Tribrid mass spectrometer with the EASY-nano LC with a 90-min chromatography gradient as previously described. Data collection and analysis were also performed as previously described.

## Functional annotation of ALDOA interactome
The identified interactome of ALDOA-WT and ALDOA-147Klac was annotated to GO terms and KEGG pathway analysis. Specifically, GO CC annotation was performed using the DAVID bioinformatics website (https://david.ncifcrf.gov/). For GO MF annotation and KEGG pathway analysis, Metascape (http://metascape.org/) was used to perform the functional enrichment analysis. For GO BP terms, Metascape was employed to perform the enrichment analysis, and Cytoscape was employed to visualize the annotation network. A group $P$ value < 0.05 for CC annotation and a group $P$ value < 0.01 for GO BP, MF annotation and KEGG pathway analysis. At least three genes were included in each group for filtering.

## Reporting summary
Further information on research design is available in the Nature Portfolio Reporting Summary linked to this article.

## Data availability
The Meltome Atlas[23], the deep proteome atlas of 29 healthy human tissues[24], the proteome of kingdoms of life[25] and an affinity-enriched lactylatome of Western flower thrips[26] used in this study were accessed through the PeoteomeXchange Consortium with the dataset identifier PXD011929, PXD010154, PXD014877, and PXD030799, respectively. Experimental data used in this study have been deposited in the ProteomeXchange Consortium and can be accessed through the Consortium via the iProX partner repository[70] with the dataset identifier PXD028488 and PXD048127. RNA-sequencing data generated in this study have been deposited in the NCBI GEO repository under accession number GSE252184. Source data are provided with this paper.

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

## Acknowledgements

This research was supported by the National Natural Science Foundation of China (grant 82104050 and 22377058 to N.W., grant 82322068 and 82173783 to H.Y., grant 82404703 to C.S.), the Natural Science Foundation of Jiangsu Province (BK20220088 to H.Y., BK20210692 to N.W., BK20241590 to C.S.), the National Key Research and Development Program of China (2021YFA1301300 to H.H.), the Fundamental Research Funds for the Central Universities (2632022YC03 to H.Y.), the Overseas Expertise Introduction Project for Discipline Innovation (G20582017001 to H.H.), Specially-appointed Professor of Jiangsu Province Project (N.W.), the supporting funds of Nanjing University of Chinese Medicine (XPT82104050 to N.W.), and the Project of State Key Laboratory of Natural Medicines, China Pharmaceutical University (SKLNMZZ202402 to H.H.).

## Author contributions

H.Y. and N.W. (Nanxi Wang) conceived the project. C.S., S.T., S.Y., R.X.T., H.H, H.Y., and N.W. (Nanxi Wang) designed the experiments. C.S., S.Y., Z.H., and N.W. (Ning Wan) performed the proteomics experiments and lactylation mining. S.T. contributed to plasmid construction. C.S., S.T., Y.Z., and Q.Y. performed the flow cytometry experiments. C.S., S.Y., S.T., C.L., and Y.Z. performed fluorescence and confocal imaging experiments. C.S., S.Y., and Y.Z. conducted metabolomics and seahorse experiments. Y.Z. performed DARTs and CETSA experiments. C.S. and Q.Y. contributed to RNA-seq and RT-qPCR analysis. C.L., Y.Z., and H.Z. contributed to co-IP analysis. S.T. and M.Z. contributed to KlacRS screening. T.W. and S.W. contributed to Klac synthesis. H.Y., C.S., and N.W. (Nanxi Wang) wrote the manuscript.

## Competing interests

The authors declare no competing interests.
