## [Transparent Peer Review file · Nature Communications]

Genetic code expansion reveals site-specific lactylation in living cells reshapes protein functions

Corresponding Author: Professor Hui Ye

Version 0:

Reviewer comments:

Reviewer #1

(Remarks to the Author)

The paper aims to develop a method to install lysine lactylation at a specific residue on a protein. This workflow enables the study of lactylation of any proteins of interest in living cells, facilitating the evaluation of perturbed activity, stability, spatial distribution, interacting proteins, and cell-wide changes caused by site-specific lactylation. The authors suggest that site-specific lactylation of ALDOA-K147 is a functional hotspot and prompts further investigation into its biological consequences. Given the importance of lysine lactylation, this paper is timely and interesting. However, the following issues should be addressed before it can be accepted.

Major Point:

1. In Figure 1b, within the cell lines, the authors investigated, what percentage of ALDOA K147 is lactylated under lactate stimulation? What is this ratio under normal culture conditions?
2. In Figure 2e, 3b, 4c, and S3a, why can only the fifth group (Figure 3b), fourth group (Figure 2e and 4c), and second line (Figure S3a) detect lactylation? Does immunoprecipitation involve in these assays? It is confused that lactylation cannot be detected in the in vivo ALDOA group and ALDOA-WT transfected group.

Reviewer #2

(Remarks to the Author)

The emphasis of wet-dry lab approach renders this manuscript very similar to your previous NatMethod 2022 publication, since you already mined different site-specific lactylation data there. Nevertheless, detailed reading revealed that you have done significant research in the functional importance of ALDOA-K147lac. Given the importance of this enzyme, I have no doubt that the manuscript has the merit for publication in NatCommun. I suggest to modify the presentation of the manuscript to focus on ALDOA-K147lac (instead of the "novel" approach). You could also remove the last part on histone lactylation (saving for the next manuscript). Alternatively, please clearly explain the novelty of this wet-dry lab approach and why it should be the highlight of the manuscript.

Apart from the presentation and structure of the manuscript, there are two technical questions. The major one is why overexpression of wt ALDOA does not fully compensate the effect of knockdown (e.g., Fig 3c/e). The lack of full compensation by wt ALDOA questions the specificity of the knockdown. Have you tried other means (e.g., different siRNA sequences, K/O, etc.)?

The other technical issue relates to the quantification of klac incorporation (Fig 3c/S2/S3):

- Fig 2c, a transfection control (e.g., 10.1093/nar/gkab132) is required. I mean simply get rid of Fig 2c and analyze samples of Fig 2d by FACS (as well as inclusion of a wt control for calculating incorporation efficiency would be fine).
- Fig S2 has the same problem as Fig 2c. Without a transfection control, the reliability of the quantification is doubtful.
- Line 218 "expression level reached approximately 40% of ALDOA-WT", Again, such a quantification/comparison is likely biased without an internal control. On the other hand, one can likely increase the level of ALDOA-147klac to that of the wt ALDOA by increasing the amount of Pyl-tRNA or other means.

Lastly, simply curious, I wonder if the chimeric PylRS has better incorporation efficiency in mammalian cells?

Reviewer #3

(Remarks to the Author)

Lactylation is a recently discovered post-translational modification (PTM). Earlier work mostly focused on histone lactylation and its role on transcriptional control. Other proteins, such as metabolic enzymes, are also heavily lactylated, but the role of lactylation on non-histone proteins are understudied. In this manuscript titled "Genetic code expansion reveals site-specific lactylation in living cells reshapes protein function", Shao, Tang, Yu et al. describes a workflow for studying lactylation on proteins in mammalian cells. This workflow is based on some of their previous work that identified lysine lactylation site from proteomics data (Reference 3), and followed by genetic code expansion that introduces site-specific lactylation to study gain of function by this PTM. Specifically, through public proteomics data mining for lactyllysine marker and affinity-enriched lactylation site identification, the authors found that K147 of ALDOA has highest occupancy among all lactylation sites in proteome. It is widely present across human tissues, and is also conserved across animal species. The authors then established an efficient system to site-specifically incorporate Klac in proteins of interest in mammalian cells. They discovered that lactylation of K147 residue blunted the aldolase activity of ALDOA, reduced glycolytic flux, and moreover, caused its nuclear localization. The relocation changes ALDOA's interactome as well as gene expression. The workflow was beautifully designed to reveal the importance of lactylation on a protein other than histones, and show the impact of PTM on enzyme function, metabolic flux, and localization. This work further highlights the elegance of genetic code expansion as a strategy in addressing the physiological importance of protein post-translational modification in cells. Overall, this work is of great importance to both the protein PTM field as well as cellular metabolism field.

We find the novelty to be in the discovery of 147KLac-induced nuclear translocation of ALDOA, as the group's previous work (Reference 3) has shown a major impact of this site-specific PTM on the enzyme activity. The impact of the work, however, could be enhanced by more evidence that reveal a functional relevance of the nuclear ALDOA, or by studying the regulatory mechanisms of this PTM. Below are a few specific comments for the authors:

1. The authors performed RNA-seq in KlacRS1/tRNA transfected cells with or without Klac. The transcriptional changes could be an indirect result of altered metabolism caused by PTM of ALDO as shown in Fig. 3, so to dissect the moonlighting function of nuclear ALDOA, controls for enzyme activity is needed. Regarding the specific experiment, is this carried on a wild type background where the enzyme function of aldolase is not affected? Was there validation on aldolase enzyme activity between the Klac+ and Klac- group?
2. The transcriptional changes point to a few molecular pathways and biological functions. Are there any observations on phenotypic changes or alteration in cellular functions? For example, this paper (doi.org/10.1038/s41467-022-35199-0) showed that nuclear localization of enzymes modulates pluripotency.
3. Regarding the functional relevance in terms of metabolic flux, the authors showed that ALDOA-Klac expression reduces the glycolytic flux (Fig. 3c). What is the evidence to rule out toxicity of the GCE system? Was the WT control also expressing the tRNA system and added Klac? Comparing Fig. 3e to Fig. 3c, there seems to be an additional drop in metabolic flux that could not be accounted for by drop in enzyme activity in the ALDO-147Klac group. Was the glycolytic flux normalized to cell number at time of assay? Some clarification or discussion is needed.
4. Further on the metabolic control by PTM, metabolic fluxes have been measured for common human cell lines (eg. DOI: [10.1126/science.1218595](https://doi.org/10.1126/science.1218595)). Does KLac occupancy show any correlation with metabolic flux? Is the Klac occupancy controlled by intracellular lactate concentration? How does it respond to lactate dehydrogenase inhibitor? It is somewhat counterintuitive to think about cancer cells being highly glycolytic while aldolase activity being strongly suppressed.

Below are some minor comments that require clarification:

1. From the proteomics analysis Fig. 1b, ALDOA-K147 is prevalent, however Fig. 3b shows almost no lactylation for WT ALDOA, why is the discrepancy?
2. Fig. 4a for nuclear localization of ALDOA, mCherry was included in the construct as well as in the image analysis, but was not mentioned in the main text. What is the purpose of mCherry? It shows nuclear enrichment which could suggest that the fluorescent reporters may have nuclear preference by their own.
3. You found that 147KLac is conserved in ALDOA from insects to mammals. Could you provide a multiple sequence alignment of all ALDOA you discussed and show the context of K147 in these proteins? Judging by the peptide MS you showed in Fig. S1C, the sequences are quite different.
4. The writing is mostly clear, but could benefit from some writing polishing service to improve clarity and avoid overclaim. For example, the first sentence of the abstract: "Still in its infancy, the functions of lactylation remain elusive". The functions of a PTM could not be in infancy as it's been evolved for millions of years. line 129, 'conservative nature' is a bad wording to describe a PTM and has bad connotations.
5. Are there any candidates for lactylation writer, reader, and eraser proteins in the interactome of ALDOA-WT and ALDOA-147Klac? Maybe some promiscuous acetylase?

Reviewer #4

(Remarks to the Author)

Version 1:

Reviewer comments:

Reviewer #1

(Remarks to the Author)

The authors have addressed my concerns.

Reviewer #2

(Remarks to the Author)

The authors have addressed my comments raised previously.

Reviewer #3

(Remarks to the Author)

The authors have provided evidence to enhance their conclusions, and nicely addressed my comments on (1) quantitation of PTM (2) better control (3) regulation by metabolic activity (4) phenotypic manifestation of gain of function. The revised manuscript is also well written. If space permits, I would suggest moving SI Fig 6c and 6d to main figure so that the statistics accompanies the image data. There are also few grammar issues that can to be corrected.

Point-by-point Response to the Reviewers' Comments

Major revisions made to this manuscript are listed as follows.

Content	Locations	Referee's comment
Assess the changes in ALDOA-147Klac levels following the regulation of lactate using a specific anti-ALDOA-147Klac antibody.	new Fig. 1g-h new SI Fig. 1f-h	Comment 1-Referee 1 Comment 4-Referee 3
Optimization of exposure parameters to visualize the relatively low levels of endogenous ALDOA-147Klac in the control and WT groups.	new Fig. 2e new Fig. 3b	Comment 2-Referee 1 Comment 5-Referee 3
Optimize the ALDOA-targeting siRNA sequences and the transfection conditions to compensate for the effect of ALDOA knockdown by overexpression of ALDOA-WT.	new Fig. 3c-e new SI Fig. 4	Comment 1-Referee 2 Comment 3-Referee 3
Use the mCherry-TAG-EGFP plasmid, which contains a built-in mCherry as an internal transfection control to quantify Klac incorporation efficiency.	new Fig. 2c	Comment 2-Referee 2
Compare Klac incorporation efficiency between KlacRS1 and the chimeric KlacRS.	new SI Fig. 2e	Comment 4-Referee 2
Assess the transcriptional changes driven by ALDOA-147Klac, rather than the altered ALDOA enzyme activity, by the following analysis: 1. Validate consistent ALDOA enzyme activity between cells that overexpress ALDOA-147Klac and those that do not by withdrawing Klac administration (the +Klac and -Klac groups). 2. Analyze ALDOA enzyme activity and transcriptional changes in cells treated with and without oxamate and FX-11, both of which will reduce lactate levels and ALDOA-147Klac expression.	new Fig. 5d new SI Fig. 7f-g	Comment 1-Referee 3
Assess phenotypic changes considering the transcriptional changes induced by ALDOA-147Klac.	new Fig. 5e-f	Comment 2-Referee 3
Perform immunofluorescence imaging experiments to validate the nuclear localization of endogenous ALDOA-147Klac.	new Fig. 4b-c	Comment 6-Referee 3
Summarize the peptide sequences containing K147 of ALDOA in the 5 analyzed species.	new SI Fig. 1c	Comment 7-Referee 3

Reviewer #1 (Remarks to the Author):

The paper aims to develop a method to install lysine lactylation at a specific residue on a protein. This workflow enables the study of lactylation of any proteins of interest in living cells, facilitating the evaluation of perturbed activity, stability, spatial distribution, interacting proteins, and cell-wide changes caused by site-specific lactylation. The authors suggest that site-specific lactylation of ALDOA-K147 is a functional hotspot and prompts further investigation into its biological consequences. Given the importance of lysine lactylation, this paper is timely and interesting. However, the following issues should be addressed before it can be accepted.

Response: We highly appreciate the reviewer's comments and have tried our best to address the issues raised by the reviewer.

Comment 1. In **Figure 1b**, within the cell lines the authors investigated, what percentage of ALDOA K147 is lactylated under lactate stimulation? What is this ratio under normal culture conditions?

Response: We appreciate the reviewer's comments. Following the reviewer's suggestion, we have generated an ALDOA-147Klac-specific antibody to directly assess the changes in ALDOA-147Klac levels following the regulation of lactate. New results are summarized as follows.

Currently, methods that accurately calculate the stoichiometry for lactylated peptides are still missing. Classic PTM site stoichiometry analysis methods (comprehensively reviewed in *Trends Biochem. Sci.*, 2019, 44, 11: 943 by Choudhary et al) mostly rely on quantifying relative abundance of modified/unmodified peptides. However, the affinity-enriched proteomics we used in **Fig. 1b** can only detect lactylated peptides. Therefore, we sought to investigate the regulatory relationship between ALDOA-147Klac occupancy and the dynamic levels of lactate by immunoblotting. An antibody that recognizes ALDOA-147Klac was generated to conveniently examine the level of ALDOA-147Klac (**Response Figure R1a**). We first incubated cells with lactate to increase intracellular lactate, and then analyzed the change in ALDOA-147Klac using the new antibody. No significant increase was observed (**Response Figure R1b**), which may be attributed to the intrinsic high level of ALDOA-147Klac. This inference is supported by its frequent detection in various cell types and organs. We then treated cells with two LDHA inhibitors, oxamate and FX-11, to reduce intracellular lactate (**Response Figure R1c**). ALDOA-147Klac was found to be significantly downregulated by this treatment (**Response Figure R1d-e**). These results suggest that ALDOA-147Klac can be regulated by modulating lactate levels.

Response Figure R1. ALDOA-147Klac occupancy can be regulated by intracellular lactate levels. (a) Dot blot assay of the generated ALDOA-147Klac antibody. Peptide 1 and 2, two synthesized ALDOA-147Klac-bearing peptides; peptide 3, an unmodified ALDOA-K147-bearing peptide. (b) Immunoblotting analysis of HEK293T cells treated with indicated concentration of sodium lactate or lactic acid. (c) Relative intensity of lactate in HEK293T cells treated with oxamate (25 mM, 24 h) or FX-11 (10 μ M, 24 h). (d) Immunoblots of ALDOA-147Klac in HEK293T cells treated with oxamate (25 mM, 24 h). (e) Immunoblots of ALDOA-147Klac in HEK293T cells treated with FX-11 (10 μ M, 24 h). For (c-e), data represent mean \pm S.D. (n=3

biologically independent samples) and the p value was calculated by unpaired two-tailed Student's t-test. **Related to Fig. 1 and Fig. S1.**

Comment 2. In **Figure 2e, 3b, 4c,** and **S3a**, why can only the fifth group (**Figure 3b**), fourth group (**Figure 2e** and **4c**), and second line (**Figure S3a**) detect lactylation? Does immunoprecipitation involve in these assays? It is confused that lactylation cannot be detected in the in vivo ALDOA group and ALDOA-WT transfected group.

Response: We highly appreciate the reviewer for raising these issues. In **Figure 2e, 3b, 4c,** and **S3a**, we performed immunoblotting experiments with a pan-lactylation antibody. The target protein ALDOA-147Klac was not immunoprecipitated, as it can be easily recognized due to its pronounced high level when introduced by the GCE method. However, endogenous lactylated proteins of similar MW but lower abundance may indeed interfere with our assessment. To overcome this challenge, we have generated an antibody against ALDOA-147Klac and believe that this helps to directly examine the levels of ALDOA-147Klac in the control group, ALDOA-WT-transfected group and ALDOA-147TAG-transfected group.

Using this anti-ALDOA-147Klac antibody, we found that the absence of ALDOA-147Klac in the control and ALDOA-WT groups is due to inappropriate exposure conditions (previous **Figure 2e, 3b, 4c,** and **S3a**). After optimizing the exposure parameters, we can visualize the relatively low, endogenous ALDOA-147Klac in the control and WT groups, and also found that the ALDOA-147TAG-transfected group produced ALDOA-147Klac at \sim 60 times and \sim 30 times higher than the endogenous and WT groups, respectively (**Response Figure R2a-b**). The relevant experiments have been re-run using the optimized exposure conditions and replaced in new **Fig. 2e** and **Fig. 3b**.

Response Figure R2. Introducing site-specific lactylation on K147 of ALDOA in living cells with genetic code expansion. (a) Immunoblotting analysis of HEK293T cells expressing endogenous ALDOA, or overexpressing Flag-tagged ALDOA-WT or ALDOA-147Klac under the indicated transfection conditions, detected against anti-ALDOA, anti-Flag and the specific anti-ALDOA-147Klac antibodies. Low exposure: 2 s; high exposure, 6 s. Data represent mean \pm S.D. (n=3 biologically independent samples) and the p value was calculated by one-way ANOVA **(b)** Immunoblots of HEK293T cells expressing ALDOA-WT or ALDOA-147Klac after knocking down the endogenous ALDOA. **Related to Fig. 2 and Fig. 3.**

b

Reviewer #2 (Remarks to the Author):

The emphasis of wet-dry lab approach renders this manuscript very similar to your previous *Nat Methods* 2022 publication, since you already mined different site-specific lactylation data there. Nevertheless, detailed reading revealed that you have done significant research in the functional importance of ALDOA-147Klac. Given the importance of this enzyme, I have no doubt that the manuscript has the merit for publication in *Nat. Commun.* I suggest modifying the presentation of the manuscript to focus on ALDOA-147Klac (instead of the "novel" approach"). You could also remove the last part on histone lactylation (saving for the next manuscript). Alternatively, please clearly explain the novelty of this wet-dry lab approach and why it should be the highlight.

Response: We highly appreciate the reviewer's insightful critiques and suggestions.

In the *Nat Methods* paper, the main aim was to establish the dry lab approach and represents a first attempt to use the GCE approach to explore the function of lactylation. However, due to the limited time available to develop the GCE system, we performed one round of positive screening, and only used purified proteins to assess the Klac-dependent change in enzyme activity. In this study, we performed further rounds of positive screening with another pyrrolysyl-tRNA synthetase (PyIRS) library and also engineered two chimeric Klac-specific tRNA synthetases (KlacRSes). Systematic evaluation of the evolved KlacRSes led to the identification of KlacRS1 as the optimal Klac-incorporation system for both *E. coli* and mammalian cells. With versatile biochemical tools, we are able to comprehensively interrogate the biological consequences induced by site-specific Klac in living cells. For ALDOA-147Klac, we found that it can induce nuclear translocation, regulate gene expression and alter PPIs, in addition to modulating the enzyme activity - the only effect shown in our *Nat Methods* study using purified proteins.

Based on these improvements, we have thoroughly revised the manuscript to highlight new advances of the GCE-based lactylation engineering system and its potential to uncover previously unknown biological significance of lactylation using ALDOA as a model protein of interest (POI). We have also removed the last part on histone lactylation as suggested. We hope that the reviewer now finds the revised manuscript suitable for publication in *Nature Communications*.

Comment 1. Apart from the presentation and structure of the manuscript, there are two technical questions. The major one is why overexpression of ALDOA-WT does not fully compensate the effect of knockdown (e.g., **Fig 3c/e**). The lack of full compensation by ALDOA-WT questions the specificity of the knockdown. Have you tried other means (e.g., different siRNA sequences, K/O, etc.)?

Response: We highly appreciate the reviewer's suggestions. Following the reviewer's suggestion, we have optimized the ALDOA-targeting siRNA sequences and the transfection conditions, and with the optimized conditions (using a different siRNA and 15 nM for transfection) we now find that ALDOA-WT rescue is able to compensate for the effect of ALDOA knockdown. As shown in **Figure R3a-c**, the enzyme activity of the ALDOA-WT-rescued group can be recovered to 85% of the control group. This has been significantly improved compared with our previous results (recovered to 50%). Similarly, metabolomics and seahorse analysis also revealed that current transfection conditions for the ALDOA-WT group allows almost full recovery of ALDOA activity following the knockdown of ALDOA (**Response Figure R3d-e**).

Response Figure R3. Lactylation on ALDOA-K147 abolished enzyme activity and regulated glycolytic flux.

(a) Crystal structure of K147 in ALDOA (PDB 4ALD) and its substrate FBP. (b) Immunoblots of HEK293T cells expressing ALDOA-WT or ALDOA-147Klac after knocking down the endogenous ALDOA. (c) ALDOA activity of HEK293T cells expressing ALDOA-WT and ALDOA-147Klac after knocking down the endogenous ALDOA. Data represent the mean \pm S.D. (n=3 biological replicates/group) and the p value was calculated by one-way ANOVA. (d) Heat map comparing abundance of metabolites involved in glycolysis. Abundance ratios were calculated by comparing ion intensities of individual metabolite in siALDOA, siALDOA+WT and siALDOA+147Klac groups, using the siCtrl group as a control. Abbreviations include: Glc, glucose; G6P, glucose 6-phosphate; FBP, fructose 1,6-bisphosphate; G3P, glycerol-3-phosphate; DHAP, dihydroxyacetone phosphate; 3PG, 3-phosphoglycerate; Pyr, pyruvate; Lac, lactate. (e) Seahorse analysis of HEK293T cells expressing ALDOA-WT or ALDOA-147Klac after knocking down the endogenous ALDOA. The ECAR was measured in real time under basal conditions and after the addition of glucose (10 mM), oligomycin (3 μ M) and 2-DG (50 mM). Left, time course of a representative experiment. Right, determination of glycolysis rate. Data represent the mean \pm S.D. (n=10 biological replicates/group) and the p value was calculated by one-way ANOVA.

Related to Fig. 3.

Comment 2. The other technical issue relates to the quantification of Klac incorporation (**Fig 2c/S2/S3**): **Fig 2c**, a transfection control (e.g., 10.1093/nar/gkab132) is required. I mean simply get rid of **Fig 2c** and analyze samples of **Fig 2d** by FACS (as well as inclusion of a WT control for calculating incorporation efficiency would be fine). **Fig S2** has the same problem as **Fig 2c**. Without a transfection control, the reliability of the quantification is doubtful.

Response: Following the reviewer's helpful suggestion, we used the mCherry-TAG-EGFP plasmid that contains a built-in mCherry as an internal transfection control. By quantifying the

proportion of mCherry+EGFP+ cells (cells with Klac incorporation) among all mCherry+ cells (total transfected cells), we can fairly assess the efficiency of Klac incorporation. In agreement with our previous results, EGFP fluorescence was detected in mCherry+ cells only after Klac administration and this ratio increased with longer incubation times, from 35.9% at 24 h to 50.2% at 48 h (**Response Figure R4a**). We have modified relevant descriptions in **Line 197-208 on Page 9** and replaced **Fig. 2c**.

We also noticed that the proportion of mCherry+ cells was consistent across the groups examined (at ~60%), indicating a consistent transfection efficiency of our experiments (**Response Figure R4a**). We believe that this consistency suggests a stable transfection efficiency of the plasmids we used. We also confirmed that all transfection experiments including those involving comparisons between WT and 147TAG groups were performed reproducibly and carefully to avoid inconsistency. Taken together, we hope that the reviewer finds it reasonable to keep the original results in **Fig. S2b and S3a** without re-performing all transfection experiments.

Response Figure R4. Incorporation efficiency of genetically encoding lactylation in living cells. Flow cytometry analysis of Klac incorporation efficiency in HEK293T cells co-transfected with the KlacRS1/tRNA^{Pyl}_{CUA} pair and mCherry-TAG-EGFP plasmids in the presence or absence of Klac (1 mM, 24 h and 48 h). Data represents the mean ± S.D. (n=3 biological replicates/group) and the p value was calculated by one-way ANOVA. **Related to Fig. 2.**

Comment 3. Line 218 "expression level reached approximately 40% of ALDOA-WT", Again, such a quantification/comparison is likely biased without an internal control. On the other hand, one can likely increase the level of ALDOA-147Klac to that of the ALDOA-WT by increasing the amount of Pyl-tRNA or other means.

Response: We apologized for the misleading description in Line 218. To compare the functional outcomes for cells expressing the site-specific lactylated ALDOA and those expressing the unmodified ALDOA, the abundance levels of overexpressed ALDOA-147Klac and ALDOA-WT should be kept consistent. As the GCE system often shows limited incorporation efficiency compared to the natural translation machinery, we can either increase the amount of the ALDOA-147TAG plasmids or reduce the amount of the ALDOA-WT plasmids to achieve comparable ALDOA expression levels. Considering excessive transfection plasmids are harmful

to cells, we chose to decrease the amount of ALDOA-WT plasmids administered. We found that treating cells with the ALDOA-WT-overexpressing plasmid at 40% of the ALDOA-147TAG plasmid resulted in equivalent ALDOA expression levels for both ALDOA-WT and ALDOA-147Klac groups. We have revised previous descriptions and discussions in **Line 233-235 on Page 10-11**.

Comment 4. Lastly, simply curious, I wonder if the chimeric PylRS has better incorporation efficiency in mammalian cells?

Response: We appreciate the reviewer's interest in the chimeric PylRS (chKlacRS-IPYE) and KlacRS systems. We have therefore evaluated the incorporation efficiency of chKlacRS-IPYE and KlacRS1 in mammalian cells and found the former to be weaker than KlacRS1, consistent with our findings in *E. coli* (**Fig. 2b**). We have added this result to **Fig. S2e (Response Figure R5)** and modified the relevant descriptions in **Line 225-228 on Page 10**.

Response Figure R5. Comparison of Klac incorporation efficiency between KlacRS1 and chKlacRS-IPYE in HEK293T cells, assessed by flow cytometry analysis using mCherry-TAG-EGFP as the reporter. **Related to Fig. S2.**

Reviewer #3 (Remarks to the Author):

Lactylation is a recently discovered post-translational modification (PTM). Earlier work mostly focused on histone lactylation and its role on transcriptional control. Other proteins, such as metabolic enzymes, are also heavily lactylated, but the role of lactylation on non-histone proteins are understudied. In this manuscript titled “Genetic code expansion reveals site-specific lactylation in living cells reshapes protein function”, Shao, Tang, Yu et al. describes a workflow for studying lactylation on proteins in mammalian cells. This workflow is based on some of their previous work that identified lysine lactylation site from proteomics data (Reference 3), and followed by genetic code expansion that introduces site-specific lactylation to study gain of function by this PTM. Specifically, through public proteomics data mining for lactyllysine marker and affinity-enriched lactylation site identification, the authors found that K147 of ALDOA has highest occupancy among all lactylation sites in proteome. It is widely present across human tissues, and is also conserved across animal species. The authors then established an efficient system to site-specifically incorporate Klac in proteins of interest in mammalian cells. They discovered that lactylation of K147 residue blunted the aldolase activity of ALDOA, reduced glycolytic flux, and moreover, caused its nuclear localization. The relocation changes ALDOA’s interactome as well as gene expression.

The workflow was beautifully designed to reveal the importance of lactylation on a protein other than histones, and show the impact of PTM on enzyme function, metabolic flux, and localization. This work further highlights the elegance of genetic code expansion as a strategy in addressing the physiological importance of protein post-translational modification in cells. Overall, this work is of great importance to both the protein PTM field as well as cellular metabolism field.

We find the novelty to be in the discovery of 147Klac-induced nuclear translocation of ALDOA, as the group’s previous work (Reference 3) has shown a major impact of this site-specific PTM on the enzyme activity. The impact of the work, however, could be enhanced by more evidence that reveal a functional relevance of the nuclear ALDOA, or by studying the regulatory mechanisms of this PTM.

Response: We highly appreciate the reviewer’s helpful critiques and suggestions and have tried our best to address the concerns raised by the reviewer. Briefly, we have validated nuclear translocation of endogenous ALDOA-147Klac in living cells using the newly generated ALDOA-147Klac-specific antibody by immunofluorescence. We have also applied different treatments to the cells and confirmed that ALDOA-147Klac, but not the alteration in ALDOA activity, led to modulation of cell adhesion. In addition, we also found phenotypic changes as the consequence of transcriptional changes induced by ALDOA-147Klac, including changes in cell morphology and cell area. We hope the revised manuscript now merits publication in *Nature Communications*.

Comment 1. The authors performed RNA-seq in KlacRS1/tRNA transfected cells with or without Klac. The transcriptional changes could be an indirect result of altered metabolism caused by PTM of ALDOA as shown in **Fig. 3**, so to dissect the moonlighting function of nuclear ALDOA, controls for enzyme activity is needed. Regarding the specific experiment, is this carried on a wild type background where the enzyme function of aldolase is not affected? Was there validation on aldolase enzyme activity between the +Klac and -Klac group?

Response: We appreciate the reviewer's insightful suggestion. We agreed that controlling ALDOA enzyme activity is crucial for dissecting its moonlighting roles in nuclear translocation. Thus, we examined the enzyme activity between the +Klac and -Klac groups. No significant difference in HEK293T cells was found (**Response Figure R6a**).

To provide more solid evidence for the moonlighting functions of nuclear ALDOA, we used two LDHA inhibitors, oxamate and FX-11, which efficiently reduced the ALDOA-147Klac level (**Response Figure R6b-c**) without affecting enzyme activity (**Response Figure R6d**). This allowed us to examine the changes in adhesion-related genes that were assigned to respond to overexpressing ALDOA-147Klac, by RT-qPCR analysis, in a WT or near WT background where the enzyme activity of ALDOA remained unaffected. We observed upregulation of pro-adhesion genes, including *CLDN1*, *LRRTM2* and *SLITRK6*, and the downregulation of an anti-adhesion gene *GPBAR1* after oxamate and FX-11 treatment (**Response Figure R6e**). The trend of changes after oxamate- and FX-11-induced downregulation of ALDOA-147Klac was opposite to that induced by GCE-enabled overexpression of ALDOA-147Klac. These findings support our conclusion that the transcriptional changes were driven by ALDOA-147Klac.

Response Figure R6. Validation of the role of ALDOA-147Klac in regulating cell adhesion.

(a) ALDOA activity of HEK293T cells co-transfected with the KlacRS1/tRNA^{Pyl}_{CUA} pair and ALDOA-147TAG plasmids in the presence or absence of Klac (5 mM, 48 h). (b) Immunoblots of ALDOA-147Klac levels in HEK293T cells treated with oxamate (25 mM, 24 h). (c) Immunoblots of ALDOA-147Klac levels in HEK293T cells treated with FX-11 (10 μM, 24 h). (d) ALDOA activity of HEK293T cells treated with or without oxamate (25 mM, 24 h) and FX-11 (10 μM, 24 h). (e) RT-qPCR analysis of *ALDOA*, *CLDN1*, *SLITRK6*, *LRRTM2* and *GPBAR1* expression levels in HEK293T cells treated with oxamate (25 mM, 24 h) or FX-11 (10 μM, 24 h). Data represent the mean ± S.D. (n=3 biological replicates/group). For (a-c), the p value was calculated by unpaired two-tailed Student's t-test. For (e), the p value was calculated by one-way ANOVA. Related to Fig. 1, Fig. S1 and Fig. 5.

Comment 2. The transcriptional changes point to a few molecular pathways and biological functions. Are there any observations on phenotypic changes or alteration in cellular functions? For example, this paper (doi.org/10.1038/s41467-022-35199-0) showed that nuclear localization of enzymes modulates pluripotency.

Response: We appreciate the reviewer's comment. Our RNA-seq analysis revealed that genes related to cell adhesion and cell junction assembly were significantly altered in the +Klac group

compared to -Klac group. qRT-PCR analysis further validated that cells expressing ALDOA-147Klac (the +Klac group) showed decreased expression of the pro-adhesion genes (*CLDN1*, *LRRTM2* and *SLITRK6*) and increased expression of the anti-adhesion gene (*GPBAR1*) (Fig. 5C), suggesting that nuclear localization of ALDOA-147Klac may modulate cell adhesion.

To assess whether cell adhesion changes were associated with morphological alterations as kindly suggested by the reviewer, we examined the cell morphology and calculated cell areas between the +Klac and -Klac groups using brightfield microscopy. Microscopic observation revealed that cells expressing ALDOA-147Klac (the +Klac group) showed reduced size, a rounded appearance, and detachment from the substrate. In contrast, cells transfected with the same plasmid but without Klac administration (the -Klac group) displayed a more spread and adherent morphology (Response Figure R7a). Accordingly, the adherent area of cells decreased significantly in the presence of Klac (Response Figure R7b). The morphology alteration is consistent with the transcriptional changes in adhesion-related genes (Fig. 5c), where ALDOA-147Klac reduced cell adhesion capacity and resulted in smaller cell areas. Collectively, our findings suggest that ALDOA-147Klac may influence cell-matrix adhesion by regulating the expression of adhesion-related genes.

Response Figure R7. Measurement of cell morphology and cell areas between the +Klac and -Klac groups. (a) Brightfield and immunofluorescence images of HEK293T cells overexpressing (the +Klac group) or not overexpressing (the -Klac group) the EGFP-tagged ALDOA-147Klac (green) due to the availability of Klac (5 mM, 48 h). The nucleus was stained with Hoechst (blue) and mCherry (red) serves as the transfection control. Scale bar, 5 μm . (b) Adhesive cell areas of cells in (a). n=40 biological replicates/group and the p value was calculated by unpaired two-tailed Student's t-test. Related to Fig. 5.

Comment 3. Regarding the functional relevance in terms of metabolic flux, the authors showed that ALDOA-Klac expression reduces the glycolytic flux (Fig. 3c). What is the evidence to rule out toxicity of the GCE system? Was the WT control also expressing the tRNA system and added Klac? Comparing Fig. 3e to Fig. 3c, there seems to be an additional drop in metabolic flux that could not be accounted for by drop in enzyme activity in the ALDO-147Klac group. Was the glycolytic flux normalized to cell number at time of assay? Some clarification or discussion is needed.

Response: We appreciate the reviewer's comments and addressed the concerns as follows.

First, regarding the toxicity of the GCE system, a previous study demonstrated that installation of GCE system into stable cell lines did not cause detectable cytotoxicity (*Nat Methods*. 2016 Feb;13(2):158-64). In addition, two other studies exploring the therapeutic potential of GCE system evaluated its toxicity in mice and found no abnormalities in the body weight or blood chemical indices (*Nat Chem Biol*. 2022 Jan;18(1):47-55 and *Nat Biomed Eng*.

2022 Feb;6(2):195-206). Based on this evidence, we believe that GCE system is safe and non-toxic.

For consistency, we also transfected the WT control cells with the K1acRS1/tRNA^{Pyl}_{CUA} pair but did not supply K1ac.

For ALDOA knockdown and overexpression experiments, we have optimized the siRNA sequence targeting ALDOA and also the transfection conditions (using a different siRNA and 15 nM for transfection). Based on the enzyme activity assay, we found that overexpression of ALDOA-WT could compensate for the effects of knockdown, with the recovery rate increased from 50% to 85% of the ctrl group. In comparison, overexpression of ALDOA-147K1ac failed to recover the ALDOA activity when compared with the ALDOA-knockdown group, suggesting dampened ALDOA activity induced by the K147-specific lactylation (**Response Figure R8a-c**). Similarly, metabolomics and seahorse analysis also showed that ALDOA-WT was able to reverse the metabolic attenuation caused by ALDOA-knockdown, whereas overexpression of ALDOA-147K1ac was unable to do so (**Response Figure R8d-e**). For the metabolomics experiments, metabolites' intensity was normalized to total ion abundance. For enzyme activity and seahorse analyses, protein concentration was used for normalization. We apologized for the previous lack of clarity and have added the relevant descriptions in **Line 1117-1118 on Page 46** and in **Line 1093-1094 on Page 45**, respectively.

Response Figure R8. Lactylation on ALDOA-K147 abolished enzyme activity and regulated glycolytic flux. (a) Crystal structure of K147 in ALDOA (PDB 4ALD) and its substrate FBP. (b) Immunoblots of HEK293T cells expressing ALDOA-WT or ALDOA-147K1ac after knocking down the endogenous ALDOA. (c) ALDOA activity of HEK293T cells expressing ALDOA-WT and ALDOA-147K1ac after knocking down the endogenous ALDOA. Data represent the mean \pm S.D. (n=3 biological replicates/group) and the p value was calculated by one-way ANOVA. (d) Heat map comparing abundance of metabolites involved in glycolysis. Abundance ratios were calculated by comparing ion intensities of individual metabolite in siALDOA, siALDOA+WT and siALDOA+147K1ac groups, using the siCtrl group as a control. Abbreviations include: Glc, glucose; G6P, glucose 6-phosphate; FBP, fructose 1,6-bisphosphate; G3P, glycerol-3-phosphate; DHAP, dihydroxyacetone phosphate; 3PG, 3-phosphoglycerate; Pyl, pyruvate; Lac, lactate. (e) Seahorse analysis of HEK293T cells expressing ALDOA-WT or ALDOA-147K1ac after knocking down the endogenous ALDOA. The ECAR was measured in real time under basal conditions and after the addition of glucose (10 mM), oligomycin (3 μ M) and 2-DG (50 mM). Left, time course of a representative experiment. Right, determination of glycolysis rate. Data represent the mean \pm S.D. (n=10 biological replicates/group) and the p value was calculated by one-way ANOVA. **Related to Fig. 3.**

Comment 4. Further on the metabolic control by PTM, metabolic fluxes have been measured for common human cell lines (eg. DOI: 10.1126/science.1218595). Does Klac occupancy show any correlation with metabolic flux? Is the Klac occupancy controlled by intracellular lactate concentration? How does it respond to lactate dehydrogenase inhibitor? It is somewhat counterintuitive to think about cancer cells being highly glycolytic while aldolase activity being strongly suppressed.

Response: We appreciate the reviewer's insightful suggestion. To elucidate the regulatory relationship between ALDOA-147Klac occupancy and changed levels of lactate, an antibody that recognizes ALDOA-147Klac was generated to directly examine the level of ALDOA-147Klac (**Response Figure R9a**). We first incubated cells with lactate to upregulate the level of intracellular lactate, and then immunoblotted ALDOA-147Klac using the new antibody. No significant increase was observed following lactate treatment (**Response Figure R9b**), which may be attributed to the intrinsic high level of ALDOA-147Klac. Interestingly, when two LDHA inhibitors, oxamate and FX-11, were administered, intracellular lactate levels were significantly reduced (**Response Figure R9c**) and ALDOA-147Klac became markedly downregulated (**Response Figure R9d-e**). These results suggest that ALDOA-147Klac can be regulated by modulating lactate levels.

Regarding the roles of ALDOA-147Klac in cancer cells, we reasoned that, as lactate can modify ALDOA and other enzymes at different sites with different occupancy, it thus remains unclear whether ALDOA-147Klac is a dominant regulator of glycolytic flux in cancer cells. Other lactylation or PTM events may activate key glycolytic enzymes and compensate for the loss of ALDOA activity, especially since ALDOA is not a rate-limiting enzyme in glycolysis (*Cell Syst.* 2018 Jul 25;7(1):49-62). In addition, our study has revealed the non-canonical functions of

ALDOA-147Klac, involving nuclear translocation and regulation of gene transcription. Considering that in this proof-of-concept study we only investigated ALDOA-147Klac-induced transcriptional changes in HEK293T cells, it is tempting to investigate the resulting changes in cancer cells in future studies to see whether ALDOA-147Klac can promote oncogenic signaling via transcription-dependent pathways. Such PTM-stimulated non-canonical functions of glycolytic enzymes have been demonstrated——acetylation at K433 of PKM2 was found to reduce the catalytic activity of PKM2, induce its translocation to the nucleus and promote the protein kinase activity for tumorigenesis (*Mol Cell*, 2013 Nov 7;52(3):340-52). We added relevant discussions to **Line 439-442 on Page20**.

Response Figure R9. ALDOA-147Klac occupancy can be regulated by intracellular lactate levels. (a) Dot blot assay of ALDOA-147Klac antibody. Peptide 1 and 2, two synthesized ALDOA-147Klac-bearing peptides; peptide 3, an unmodified ALDOA-K147-bearing peptide. (b) Immunoblotting analysis of HEK293T cells treated with indicated concentration of sodium lactate or lactic acid. (c) Relative intensity of lactate in HEK293T cells treated with oxamate (25 mM, 24h) or FX-11 (10 μ M, 24 h). (d) Immunoblots of ALDOA-147Klac levels in HEK293T cells treated with oxamate (25 mM, 24 h). (e) Immunoblots of ALDOA-147Klac levels in HEK293T cells treated with FX-11 (10 μ M, 24 h). For (c-e), data represent the mean \pm S.D. (n=3 biologically independent samples) and the p value was calculated by unpaired two-tailed Student's t-test. **Related to Fig. 1 and Fig. S1.**

Below are some minor comments that require clarification:

Comment 5. From the proteomics analysis **Fig. 1b**, ALDOA-K147 is prevalent, however **Fig. 3b** shows almost no lactylation for WT ALDOA, why is the discrepancy?

Response: We highly appreciate the reviewer for raising these issues. In **Fig. 3b**, we performed immunoblotting experiments with a pan-lactylation antibody. The target protein ALDOA-147Klac was not immunoprecipitated, as it can be easily recognized due to its pronounced high level when introduced by the GCE method. However, endogenous lactylated proteins of similar MW but lower abundance may indeed interfere with our assessment. To overcome this challenge, we have generated an antibody against ALDOA-147Klac and believe that this helps to directly examine the levels of ALDOA-147Klac in the control group, ALDOA-WT-transfected group and

ALDOA-147TAG transfected group.

Using this anti-ALDOA-147Klac antibody, we found that the absence of ALDOA-147Klac in the control and ALDOA-WT groups is due to inappropriate exposure conditions (previous **Fig. 3b**). After changing the exposure parameters, we can visualize the relatively low, endogenous ALDOA-147Klac in the control and WT groups, and also found that the ALDOA-147TAG-transfected group produced ALDOA-147Klac at ~60 times and ~30 times higher than the endogenous and WT groups, respectively (**Response Figure R10a-b**). The relevant experiments have all been re-run using the optimized exposure conditions and replaced by a new **Fig. 2e** and **Fig. 3b**.

Response Figure R10. Introducing site-specific lactylation on K147 of ALDOA in living cells with genetic code expansion. (a) Immunoblotting analysis of HEK293T cells expressing endogenous ALDOA, or overexpressing Flag-tagged ALDOA-WT or ALDOA-147Klac under the indicated transfection conditions, detected against anti-ALDOA, anti-Flag and the specific anti-ALDOA-147Klac antibodies. Low exposure: 2 s; high exposure, 6 s. Data represent mean \pm S.D. (n=3 biologically independent samples) and the p value was calculated by one-way ANOVA **(b)** Immunoblots of HEK293T cells expressing ALDOA-WT or ALDOA-147Klac after knocking down the endogenous ALDOA. **Related to Fig. 2 and Fig. 3.**

Comment 6. Fig. 4a for nuclear localization of ALDOA, mCherry was included in the construct as well as in the image analysis, but was not mentioned in the main text. What is the purpose of mCherry? It shows nuclear enrichment which could suggest that the fluorescent reporters may have nuclear preference by their own.

Response: We apologize for the missing information about mCherry in the main text. Briefly, the plasmids mCherry-T2A-ALDOA (147TAG)-EGFP and mCherry-T2A-ALDOA (WT)-EGFP were designed and used in **Fig. 4a**. The T2A element allows for the co-expression of mCherry and ALDOA-EGFP without interfering with each other in living cells. As a result, mCherry is released as a separate internal transfection control to ensure the ALDOA-WT and ALDOA-147Klac groups have been transfected with the same amount of plasmid as suggested by Reviewer 2. The introduction of mCherry is also used to identify the successfully transfected cells for subsequent analysis of ALDOA-EGFP from the mCherry-positive cells. Then, ALDOA-EGFP is used to visualize and quantify the subcellular partitioning of ALDOA-WT and ALDOA-147Klac. We have added relevant descriptions in **Line 315-322 on Page 14**.

To confirm that endogenous ALDOA-147Klac is also enriched in the nucleus, we generated an anti-ALDOA-147Klac antibody to directly visualize the localization of ALDOA-147Klac in living cells without relying on EGFP and the GCE approach. Immunofluorescence analysis supports our previous findings that ALDOA-147Klac was enriched in the nucleus, whereas

ALDOA was mainly located in the cytoplasm (**Response Figure R11a, ctrl group**). Furthermore, both nuclear and total levels of ALDOA-147Klac significantly decreased when lactate was reduced by treating cells with two LDHA inhibitors, oxamate and FX-11, respectively (**Response Figure R11b**). Therefore, both GCE-based engineered and endogenous ALDOA-147Klac tend to localize to the nucleus.

Response Figure R11. Subcellular localization of ALDOA and ALDOA-147Klac. (a) Immunofluorescence imaging of ALDOA and ALDOA-147Klac analyzed by confocal microscopy using cells treated without and with oxamate (25 mM, 24 h) or FX-11 (10 μ M, 24 h). Left, HEK293T cells co-stained with the anti-ALDOA (red) and anti-ALDOA-147Klac antibodies (green) and Hoechst (Blue). Right, fluorescence intensity profiles across the indicated lines in the left. Scale bar, 20 μ m. (b) Analysis of the mean fluorescence intensity of whole-cell and nuclear ALDOA-147Klac for cells in (a). n=69 for the control group, n=68 for the oxamate-treated group and n=65 for the FX-11-treated group. The p value was calculated by one-way ANOVA. **Related to Fig. 4.**

Comment 7. You found that 147Klac is conserved in ALDOA from insects to mammals. Could you provide a multiple sequence alignment of all ALDOA you discussed and show the context of K147 in these proteins? Judging by the peptide MS you showed in **Fig. S1C**, the sequences are quite different.

Response: We highly appreciate the reviewer's suggestion. We have aligned the peptide sequences containing K147 of ALDOA in *Oryctolagus cuniculus*, *Homo sapiens*, *Mus musculus*, *Frankliniella occidentalis* and *Drosophila melanogaster*, and found that this K147 site is highly conserved in these species. We have summarized their sequences in **Response Figure R12** and added the corresponding descriptions in **Line 146-149 on Page 6-7**.

Response Figure R12. Evolutionary conservation of K147 on ALDOA across eukaryotic phylogeny. **Related to Fig. S1C.**

ALDOA Sequence

Homo Sapiens	...KDGADFAK W RRCVLKIG...
	140 147 155
Oryctolagus cuniculus	...KDGADFAK W RRCVLKIG...
Mus musculus	...KDGADFAK W RRCVLKIG...
Drosophila melanogaster	...KDGCDFAK W RRCVLKIG...
Frankliniella occidentalis	...KDGCHFAK W RRCVLKIG...

Comment 8. The writing is mostly clear, but could benefit from some writing polishing service to improve clarity and avoid overclaim. For example, the first sentence of the abstract: “Still in its infancy, the functions of lactylation remain elusive”. The functions of a PTM could not be in infancy as it’s been evolved for millions of years. line 129, ‘conservative nature’ is a bad wording to describe a PTM and has bad connotations.

Response: We appreciate the reviewer’s comment and have thoroughly revised the manuscript. We have also shortened the manuscript to improve clarity.

Comment 9. Are there any candidates for lactylation writer, reader, and eraser proteins in the interactome of ALDOA-WT and ALDOA-147Klac? Maybe some promiscuous acetylase?

Response: We highly appreciate the reviewer’s insightful critiques and suggestions. In this study, we used Co-IP to map the interactome of ALDOA-WT and ALDOA-147Klac, which is effective for identifying high-affinity and stable protein interactions. However, protein-protein interactions between PTM-modified proteins and their writers, erasers and readers are usually weak and transient. Photo-crosslinking-based techniques are more suitable for identifying such interactions, as they can convert weak and transient interactions between the PTM-modified proteins and their writer/eraser/reader into covalent binding partners (*Nat Chem Biol.* 2016 Feb;12(2):70-2, *Science.* 2023 Feb 17;379(6633):717-723 and *Nat Commun.* 2024 Feb 17;15(1):1465). Although we are actively working on identifying the writer/eraser/reader of ALDOA-147Klac, this investigation is beyond the scope of the current manuscript. We have added relevant discussions to **Line 467-475 on Page 21** and **references [61-63]**.

Point-by-point Response to the Reviewers' Comments

Reviewer #1 (Remarks to the Author):

The authors have addressed my concerns.

Response: We sincerely appreciate the reviewer's comments and dedicated effort in reviewing our manuscript.

Reviewer #2 (Remarks to the Author):

The authors have addressed my comments raised previously.

Response: We highly appreciate the reviewer's insightful suggestions and great efforts in reviewing our manuscript.

Reviewer #3 (Remarks to the Author):

The authors have provided evidence to enhance their conclusions, and nicely addressed my comments on (1) quantitation of PTM (2) better control (3) regulation by metabolic activity (4) phenotypic manifestation of gain of function. The revised manuscript is also well written. If space permits, I would suggest moving SI Fig 6c and 6d to main figure so that the statistics accompanies the image data. There are also few grammar issues that can be corrected.

Response: We highly appreciate the reviewer's helpful critiques and suggestions. We have moved Fig. S6c and S6d to Fig. 4 as new Fig. 4b and 4c, respectively. We also sincerely apologize for the grammar mistakes and have made the corrections throughout the manuscript.